# Conformal Prediction: A Theoretical Note and Benchmarking Transductive Node Classification in Graphs

**Pranav Maneriker**[*][†]                                    *pmane@dolby.com*
*Dolby Laboratories*

**Aditya T. Vadlamani**[*]                                    *vadlamani.12@osu.edu*
*Department of Computer Science and Engineering*
*The Ohio State University*

**Anutam Srinivasan**                                    *srinivasan.268@osu.edu*
*Department of Electrical and Computer Engineering*
*The Ohio State University*

**Yuntian He**[†]                                    *heyuntian.cn@gmail.com*
*Meta*

**Ali Payani**                                    *apayani@cisco.com*
*Cisco Systems*

**Srinivasan Parthasarathy**                                    *srini@cse.ohio-state.edu*
*Department of Computer Science and Engineering*
*The Ohio State University*

**Reviewed on OpenReview:** *https://openreview.net/forum?id=Ed1DBB3sBQ*

## Abstract

Conformal prediction has become increasingly popular for quantifying the uncertainty associated with machine learning models. Recent work in graph uncertainty quantification has built upon this approach for conformal graph prediction. The nascent nature of these explorations has led to conflicting choices for implementations, baselines, and method evaluation. In this work, we analyze the design choices made in the literature and discuss the tradeoffs associated with existing methods. Building on the existing implementations, we introduce techniques to scale existing methods to large-scale graph datasets without sacrificing performance. Our theoretical and empirical results justify our recommendations for future scholarship in graph conformal prediction.

## 1 Introduction

Modern machine learning models trained on losses based on point predictions are prone to be overconfident in their predictions (Guo et al., 2017). The Conformal Prediction (CP) framework (Vovk et al., 2005) provides a mechanism for generating statistically sound post hoc prediction sets (or intervals, in case of continuous outcomes) with coverage guarantees under mild assumptions. Usually, the assumption made in CP is that the data is exchangeable, i.e., the joint distribution of the data is invariant to permutations of the data points. CP's guarantees are distribution-free and can be added post hoc to arbitrary black-box predictor scores, making them ideal candidates for quantifying uncertainty in complex models, such as neural networks.

Network-structured data, such as social, transportation, and biological networks, is ubiquitous in modern data science applications. Graph Neural Networks (GNNs) have been developed to model vector representations

---

[*]Equal Contribution. [†]This work was done while the author was a student at The Ohio State University

of network-structured data and be effective in a variety of tasks such as node classification, link prediction, and graph classification (Hamilton, 2020; Wu et al., 2022). As the network structure introduces possible dependencies between the data points, uncertainty quantification approaches built for independent and identically distributed (iid) data cannot be directly applied to graph data. However, recent work (Clarkson, 2023; Zargarbashi et al., 2023; Huang et al., 2023) has demonstrated that, in specific settings, CP can be applied to graph data to generate statistically sound prediction sets for the node classification task, similar to other classification tasks. These methods were designed to leverage the graph structure and homophily to improve upon baseline non-graph CP methods. Variations of CP, ranging from most to least computationally expensive, include (1) **Full-CP** (Vovk et al., 2005), (2) **Cross-CP/CV+/Jackknife+** (Vovk, 2015; Barber et al., 2021), and (3) **Split (or Inductive)-CP** (Papadopoulos et al., 2002; Papadopoulos, 2008). Prior work on graph CP has mainly focused on the Split-CP setting due to its computational efficiency and the distribution-free guarantees with black-box models. Thus, we also focus on Split-CP for our work.

**Key Contributions:** This work makes the following contributions:

1. We perform a comprehensive analysis of the existing graph split-CP literature (e.g., CFGNN, DAPS) for transductive node classification. This analysis aims to understand the choices made in method implementation (i.e., scalability) and method evaluation (i.e., dataset preparation and baseline selection). We also extend this analysis to include an analysis of when these graph-based approaches improve over non-graph CP methods, such as TPS and APS.

2. We present a theorem for evaluating the efficiency (set size) of different conformal prediction methods *prior to* deployment.

3. We develop a Python library that implements our analysis, enabling practitioners to efficiently apply and evaluate conformal prediction for graph data, with scalability enhancements for large graphs, particularly for compute-intensive methods like CFGNN.

## 2 Conformal Prediction

Conformal prediction is used to quantify model uncertainty by providing prediction sets/intervals with rigorous coverage guarantees. We will focus on conformal prediction in the classification setting. Given a calibration dataset $\mathcal{D}_{\text{calib}} = \{(\boldsymbol{x}_i, y_i)\}_{i=1}^n$, where $\boldsymbol{x}_i \in \mathcal{X} = \mathbb{R}^d$ and $y_i \in \mathcal{Y} = \{1, \ldots, K\}$, conformal prediction can be used to construct a prediction set $\mathcal{C}$ for an unseen test point $(\boldsymbol{x}_{n+1}, y_{n+1})$ such that

$$\Pr\left[y_{n+1} \in \mathcal{C}(\boldsymbol{x}_{n+1})\right] \geq 1 - \alpha$$

where $1 - \alpha \in (0, 1)$ is a user-specified coverage level. The only assumption required for the coverage guarantee is that $\mathcal{D}_{\text{calib}} \cup \{(\boldsymbol{x}_{n+1}, y_{n+1})\}$ is exchangeable. The following theorem provides a general recipe for constructing a prediction set with a coverage guarantee.

**Theorem 2.1** ((Vovk et al., 2005; Angelopoulos & Bates, 2023)). *Suppose $\{(\boldsymbol{x}_i, y_i)\}_{i=1}^{n+1}$ are exchangeable, $s : \mathcal{X} \times \mathcal{Y} \to \mathbb{R}$ is a score function measuring the non-conformity of point $(\boldsymbol{x}, y)$, with higher scores indicating lower conformity, and a target miscoverage level $\alpha \in [0, 1]$. Let $\hat{q}(\alpha) = \text{Quantile}\left(\frac{\lceil (n+1)(1-\alpha) \rceil}{n}; \{s(\boldsymbol{x}_i, y_i)\}_{i=1}^n\right)$ be the conformal quantile. Define $\mathcal{C}(\boldsymbol{x}) = \{y \in \mathcal{Y} : s(\boldsymbol{x}, y) \leq \hat{q}(\alpha)\}$. Then,*

$$1 - \alpha + \frac{1}{n+1} \geq \Pr\left[y_{n+1} \in \mathcal{C}(\boldsymbol{x}_{n+1})\right] \geq 1 - \alpha \tag{1}$$

*The upper bound assumes that the scores are unique or that a suitably random tie-breaking method exists.*

The function $s$ is the non-conformity score function, and it measures the degree of non-agreement between an input $\boldsymbol{x}$ and the label $y$, given exchangeability with the calibration data $\mathcal{D}_{\text{calib}}$, i.e., larger scores indicate worse agreement between $\boldsymbol{x}$ and $y$. While Theorem 2.1 does not restrict the choice of $s$, this choice can significantly impact the size of the prediction set.

The setup of Theorem 2.1 is called Split-CP, as the score function remains fixed for the calibration split. In other versions of CP, the score function is usually more expensive to compute as it requires some

$1 < k \leq n$ calibration points in addition to the unseen test point. For Full-CP, we have $k = n$, whereas for Cross-CP/CV+/Jackknife+, $k < n$. When applying Split-CP, the dataset is partitioned into $\mathcal{D} = \mathcal{D}_{\text{train}} \cup \mathcal{D}_{\text{valid}} \cup \mathcal{D}_{\text{calib}} \cup \mathcal{D}_{\text{test}}$.

## 2.1 Conformal Prediction in Graphs

The usual tasks of interest with graph data include node classification, link prediction, and graph classification. This work focuses on node classification and its extensions to conformal prediction. Consider an attributed homogeneous graph $\mathcal{G} = (\mathcal{V}, \mathcal{E}, \boldsymbol{X})$, where $\mathcal{V}$ is the set of nodes, $\mathcal{E}$ is the set of edges, and $\boldsymbol{X}$ is the set of node attributes. Let $\boldsymbol{A}$ denote the adjacency matrix for the graph. Further, let $\mathcal{Y} = \{1, \ldots, K\}$ denote the set of class labels associated with the nodes. For $v \in \mathcal{V}$, $\boldsymbol{x}_v \in \mathcal{X} = \mathbb{R}^d$ denotes its features and $y_v \in \mathcal{Y}$ denotes its true class label. The task of node classification is to learn a model that predicts the label for each node given node features and the adjacency matrix, i.e., $(\boldsymbol{X}, \boldsymbol{A}, v) \mapsto y_v$. Corresponding to the CP partitions, we denote the nodes in the training set as $\mathcal{V}_{\text{train}}$, the validation set as $\mathcal{V}_{\text{valid}}$, the calibration set as $\mathcal{V}_{\text{calib}}$, and the test set as $\mathcal{V}_{\text{test}}$. We denote $\mathcal{V}_{\text{dev}} = \mathcal{V}_{\text{train}} \cup \mathcal{V}_{\text{valid}}$ as the development set for the base model. Note that labels are available only for nodes in $\mathcal{V}_{\text{train}}$, $\mathcal{V}_{\text{valid}}$, and $\mathcal{V}_{\text{calib}}$, and must be predicted for nodes in $\mathcal{V}_{\text{test}}$. The model cycle will involve four phases, viz. training, validation, calibration, and testing. Next, we discuss the different settings for node classification in graphs and the applicability of conformal prediction.

**Transductive setting** In this setting, the model has access to the fixed graph $\mathcal{G}$ during the model cycle. The nodes used in the model cycle are split into $\mathcal{V}_{\text{train}}$, $\mathcal{V}_{\text{valid}}$, and $\mathcal{V}_{\text{calib}} \cup \mathcal{V}_{\text{test}}$. The specific $\mathcal{V}_{\text{calib}}$ and $\mathcal{V}_{\text{test}}$ are subsets of $\mathcal{V}_{\text{calib}} \cup \mathcal{V}_{\text{test}}$. This is the setting considered by Zargarbashi et al. (2023) and Huang et al. (2023). Note that the labels for $\mathcal{V}_{\text{calib}}$ are not available for training and validation of the base model; however, all the neighborhood information of $\mathcal{G}$ and the features $\boldsymbol{x}_v$ and labels $y_v$, for $v \in \mathcal{V}_{\text{dev}}$ are available. During the calibration phase, the $(\boldsymbol{x}_v, y_v)$ for $v \in \mathcal{V}_{\text{calib}}$ and all the neighborhood information are used to compute the non-conformity scores. This split ensures that the base model cannot distinguish between the calibration and test nodes, and hence exchangeability holds for $v \in \mathcal{V}_{\text{calib}} \cup \mathcal{V}_{\text{test}}$.

**Inductive setting** In this setting, the model is only trained and calibrated on a fixed graph $\mathcal{G}$ during the model cycle. Once the calibration phase is complete, $\mathcal{G}$ changes to include new unseen test points, breaking the exchangeability assumption as nodes arriving later in the sequence will have access to neighbors that came earlier. This setting is infrequently studied as it requires making specific assumptions about graph structure and the data generation process (Clarkson, 2023; Zargarbashi & Bojchevski, 2024).

In line with previous work, we focus on the ***transductive*** setting. The following theorem states that a scoring model trained on the calibration set will generate scores exchangeable with the test set in the transductive setting, thus allowing for conformal prediction.

**Theorem 2.2** ((Zargarbashi et al., 2023; Huang et al., 2023))**.** *Let* $\mathcal{G} = (\mathcal{V}, \mathcal{E}, \boldsymbol{X})$ *be an attributed graph, and* $\mathcal{V}_{calib} \cup \mathcal{V}_{test}$ *be exchangeable. Let* $F : \mathcal{X}^{|V|} \to \Delta^{|V| \times K}$ *be any permutation equivariant model on the graph (e.g., GNNs). Define* $F(G) = \Pi \in \Delta^{|V| \times K}$ *as the output probability matrix for a model trained on only* $\mathcal{V}_d$. *Then any score function* $s(v, y) = s(\Pi_v, y, \mathcal{G})$ *is exchangeable for all* $v \in \mathcal{V}_{calib} \cup \mathcal{V}_{test}$.

The intuition for this theorem is that if the underlying model does not depend on the order of the nodes in the graph, then the outputs will also be exchangeable. This holds for most standard GNNs. The formal proof for this theorem is available in (Zargarbashi et al., 2023; Huang et al., 2023). This theorem paves the way for using conformal prediction for transductive node classification.

For the following sections, we will assume that the base model $\hat{\pi} : \mathcal{X} \to \Delta_{\mathcal{Y}}$, where $\Delta_{\mathcal{Y}}$ is the probability simplex over $\mathcal{Y}$, is learned using the training and validation sets $\mathcal{D}_{\text{train}} \cup \mathcal{D}_{\text{valid}}$. The calibration set $\mathcal{D}_{\text{calib}}$ is used to determine the $\hat{q}(\alpha)$ from Theorem 2.1, and the test set $\mathcal{D}_{\text{test}}$ is the set for which we want to construct our prediction sets. Note that $\mathcal{D}_x = \{(\boldsymbol{x}_v, y_v) \mid v \in \mathcal{V}_x\}$ for graph datasets. Note that the scores need not lie over a simplex, but instead be in $\mathbb{R}^K$. However, this greatly simplifies the exposition for the following sections and is standard practice in prior work.

## 2.2 Conformal Prediction Methods

Several conformal prediction methods are discussed in the CP for transductive node classification literature, including the following general and graph-specific methods.

### 2.2.1 General Methods Applied to Transductive Node Classification

**Threshold Prediction Sets (TPS) (Sadinle et al., 2019)** In TPS the score function is $s(\boldsymbol{x}, y) = 1 - \hat{\pi}(\boldsymbol{x})_y$, where $\hat{\pi}(\boldsymbol{x})_y$ is the class probability for the correct class. This is the simplest method, which is also shown to be optimal with respect to efficiency, thus making it essential to include as a baseline. However, TPS is known to achieve this efficiency by under-covering hard examples and over-covering easy examples (Angelopoulos et al., 2021; Zargarbashi et al., 2023). We note that this discrepancy is claimed to occur as the TPS scores are not *adaptive*, and consider only one dimension of the score for each calibration sample. However, Sadinle et al. (2019) also proposed a classwise controlled version of TPS. Instead of defining a single threshold for all classes, they separately compute the threshold for each class for a corresponding $\alpha$. Thus, we define classwise quantile thresholds for each $y \in \mathcal{Y}$

$$\hat{q}(\alpha_y, y) = \text{Quantile}\left(\frac{\lceil (n+1)(1 - \alpha_y) \rceil}{n}; \{s(\boldsymbol{x}_i, y_i) \mid i = 1, \ldots, n, y_i = y\}\right) \tag{2}$$

and the corresponding prediction sets as

$$\mathcal{C}(\boldsymbol{x}) = \{y \in \mathcal{Y} : s(\boldsymbol{x}, y) \leq \hat{q}(\alpha_y, y)\}.$$

Note that this version achieves coverage for each class label, making it more adaptive (i.e., $\Pr(y_{n+1} \in \mathcal{C}(\boldsymbol{x}) \mid y_{n+1} = y) \geq 1 - \alpha, \ \forall y \in \mathcal{Y}$). The version defined by Sadinle et al. (2019) allows controlling $\alpha_y$ for each class, though we set $\alpha_y = \alpha$ for class-adaptability. The trade-off with the adaptive version is that we have fewer calibration samples for each quantile threshold dimension, which may lead to higher variance in the distribution of coverage (Vovk, 2012). We refer to this variation **TPS-Classwise**.

**Adaptive Prediction Sets (APS) (Romano et al., 2020)** APS is the most popular CP baseline for classification tasks. For APS, the softmax logits are sorted in descending order, and the score is the cumulative sum of the class probabilities until the correct class. For tighter prediction sets, randomization can be introduced through a uniform random variable. Formally, if $\hat{\pi}(\boldsymbol{x})_{(1)} \geq \hat{\pi}(\boldsymbol{x})_{(2)} \geq \cdots \geq \hat{\pi}(\boldsymbol{x})_{(K-1)}$, $u \sim U(0, 1)$, and $r_y$ is the rank of the correct label, then

$$s(\boldsymbol{x}, y) = \left[\sum_{i=1}^{r_y} \hat{\pi}(\boldsymbol{x})_{(i)}\right] \underbrace{- u\hat{\pi}(\boldsymbol{x})_y}_{\text{rand. term}}.$$

**Regularized Adaptive Prediction Sets (RAPS) (Angelopoulos et al., 2021)** One drawback of APS is that it can produce large prediction sets. RAPS incorporates a regularization term for APS. Given the same setup as APS, define $o(\boldsymbol{x}, y) = |\{c \in \mathcal{Y} : \hat{\pi}(\boldsymbol{x})_y \geq \hat{\pi}(\boldsymbol{x})_c\}|$. Then,

$$s(\boldsymbol{x}, y) = \left[\sum_{i=1}^{r_y} \hat{\pi}(\boldsymbol{x})_{(i)}\right] - u\hat{\pi}(\boldsymbol{x})_y + \nu \cdot \max\{(o(\boldsymbol{x}, y) - k_{reg}), 0\},$$

where $\nu$ and $k_{reg} \geq 0$ are regularization hyperparameters.

### 2.2.2 Graph Methods

Graphs are rich in structure and network homophily, and recent works in graph CP leverage these properties when developing graph-specific methods. Intuitively, network homophily suggests that non-conformity scores and efficiencies of connected nodes are related.

**Diffusion Adaptive Prediction Sets (DAPS) (Zargarbashi et al., 2023)** To leverage network homophily, DAPS performs a one-step diffusion update on the non-conformity scores. Formally, if $s(\boldsymbol{x}, y)$ is a point-wise score function, then the diffusion step gives a new score function

$$\hat{s}(\boldsymbol{x}, y) = (1 - \delta)s(\boldsymbol{x}, y) + \frac{\delta}{|\mathcal{N}_{\boldsymbol{x}}|} \sum_{\boldsymbol{u} \in \mathcal{N}_{\boldsymbol{x}}} s(\boldsymbol{u}, y),$$

where $\delta \in [0, 1]$ is a diffusion hyperparamter and $\mathcal{N}_{\boldsymbol{x}}$ is the 1-hop neighborhood of $\boldsymbol{x}$. More details and results on DAPS can be found in Appendix C.1.

**Diffused TPS-Classwise (DTPS)** In the case of DAPS, $s$ is chosen to be APS; however, any adaptive score function can be used. In this work, we consider applying diffusion to the scores from TPS-Classwise (the adaptive variation of TPS). More details and results on DTPS can be found in Appendix C.1.

**Neighborhood Adaptive Prediction Sets (NAPS) (Clarkson, 2023)** NAPS leverages the neighborhood information of a test point to compute a weighted conformal quantile for constructing the prediction set. It is the only method in the study that can work with non-exchangeable data, but it loosens the standard bounds of conformal prediction. The weighting functions used with NAPS are uniform, hyperbolic, and exponential. A detailed explanation of NAPS, the weighting functions, and how it is adapted to the transductive setting is in Appendix C.2.

**Conformalized GNN (CFGNN) (Huang et al., 2023)** CFGNN is a model-based approach to graph CP. Similar to ConfTr (Stutz et al., 2021), CFGNN uses a second model and an inefficiency-based loss function to correct the scores from the base model. This is feasible as all the steps of the conformal prediction framework (i.e., non-conformity score computation, quantile computation, thresholding) can be expressed as differentiable operations (Stutz et al., 2021). The underlying observation for CFGNN is that the inefficiencies are correlated between nodes with similar neighborhood topologies, hence, the secondary model used is a GNN. The inefficiency loss includes a point-wise score function that differs for training and validation. In our work, we set the score function to be APS with randomization between training and validation. More details can be found in Appendix C.3.

## 2.3 A Theoretical Note on Randomization.

Recall that APS (Romano et al., 2020) can be implemented in both a randomized and a deterministic manner (see Appendix A for derivations), by using a uniform random variable to determine if the correct class is included or not in the prediction set, during the calibration phase. Both versions of APS provide conditional coverage guarantees; however, the deterministic version has a simpler exposition, so this version is implemented in the popular monographs on conformal prediction by Angelopoulos & Bates (2023). However, the lack of randomization may result in larger prediction sets. This modification affects computing the conformal quantile during the calibration phase and the prediction set construction during the test phase.

We will formally present the conditions that impact (prediction) efficiency, and later apply it to APS with and without randomization. Let $A(\boldsymbol{x}, y)$ be any non-conformity score function, and let,

$$\hat{q}_A = \text{Quantile}\left(\frac{\lceil (n+1)(1-\alpha) \rceil}{n}; \{A(\boldsymbol{x}_i, y_i)\}_{i=1}^n\right).$$

Consider the exchangeable sequence, $\left\{A(\boldsymbol{x}_i, y_i^R)\right\}_{i=1}^{n+1}$, where $y_i^R$ is a randomly selected incorrect label[1]. We now define $\alpha_c^A \in [0, 1]$ such that:

$$\hat{q}_A = \text{Quantile}\left(\frac{\lceil (n+1)(1-\alpha_c^A) \rceil}{n}; \{A(\boldsymbol{x}_i, y_i^R)\}_{i=1}^n\right). \tag{3}$$

---

[1]For an explanation on why this is exchangeable see Appendix B

Table 1: Summary statistics for datasets. Predefined splits from the original sources are noted.

| Dataset | Nodes | Edges | Classes | Features | # Train | # Valid | # Test |
|---------|-------|-------|---------|----------|---------|---------|--------|
| Amazon_Computers | 13,752 | 491,722 | 10 | 767 | - | - | - |
| Cora | 19,793 | 126,842 | 70 | 8,710 | - | - | - |
| Coauthor_CS | 18,333 | 163,788 | 15 | 6,805 | - | - | - |
| Flickr | 89,250 | 899,756 | 7 | 500 | 44,625 | 22,312 | 22,313 |
| ogbn-arxiv | 169,343 | 1,166,243 | 40 | 128 | 90,941 | 29,799 | 48,603 |

In other words, $\alpha_c^A$ is the miscoverage level for the random incorrect label when using $\hat{q}_A$ to construct the prediction set. By leveraging a recent result in the literature (Vadlamani et al., 2025), we can compute $\alpha_c^A$ and see that the following result holds (see Appendix B).

$$1 - \alpha_c^A \leq \Pr[y_{n+1}^R \in \mathcal{C}_A(\boldsymbol{x}_{n+1})] \leq 1 - \alpha_c^A + \frac{1}{n+1}. \tag{4}$$

Let $\tilde{A}(\boldsymbol{x}, y)$ be another non-conformity score and define $\{\tilde{A}(\boldsymbol{x}_i, y_i^R)\}_{i=1}^{n+1}$ and $\alpha_c^{\tilde{A}}$ similarly. The following theorem gives a sufficient condition for when $A$ is more efficient than $\tilde{A}$.

**Theorem 2.3.** *If $\alpha_c^A - \alpha_c^{\tilde{A}} \geq \frac{2}{(n+1)}$ then score function $A$ produces a more efficient prediction set than $\tilde{A}$. Formally, $\mathbb{E}\left[|\mathcal{C}_{\tilde{A}}(\boldsymbol{x}_{n+1})| - |\mathcal{C}_A(\boldsymbol{x}_{n+1})|\right] \geq 0$*

With Theorem 2.3, we can further determine the expected efficiency improvement as the calibration set size gets arbitrarily large, as seen in the following corollary. Proofs for both results are available in Appendix B.

**Corollary 2.3.1.** *As $n \to \infty$, $\mathbb{E}[|\mathcal{C}_{\tilde{A}}(\boldsymbol{x}_{n+1})| - |\mathcal{C}_A(\boldsymbol{x}_{n+1})|] = (K-1)\left(\alpha_c^A - \alpha_c^{\tilde{A}}\right)$*

Corollary 2.3.1 allows practitioners to gauge the expected efficiency (i.e., set size) benefits prior to the inference phase.

## 3 Empirical Analysis and Insights

**Key Research Questions:** In the subsequent sections, this work will answer the following key questions raised when evaluating the literature on CP for transductive node classification.

1. TPS-Classwise is largely overlooked in the recent literature, given the popularity of APS and TPS not being adaptive. Is TPS-Classwise a viable baseline? (Discussed in Section 3.1).

2. Theorem 2.3 provides a framework for evaluating when a method is more efficient (that is, produces smaller sets) than another. What are the empirically observed implications of Theorem 2.3 on APS? (Discussed in Sections 3.2 and 3.3).

3. Can we scale the computationally intensive CFGNN to handle large-scale graphs, such as ogbn-arxiv, and at what cost to efficiency and coverage, if any? (Discussed in Sections 3.3 and 3.4).

4. Given the theoretical, experimental, and methodological analysis, what are some key insights and guidelines for future scholarship in this area? (Discussed in Sections 3.5).

**Datasets.** Table 1 contains a representative set of datasets of varying sizes (i.e., number of nodes/edges) and the number of classes evaluated in this section. The Appendix contains the list of all the datasets used in this study. We used the dataset versions available on the Deep Graph Library (Wang et al., 2019) and Open Graph Benchmark (Hu et al., 2020).

**Methods.** Section 2.1 includes all of the baseline methods considered in the experiments.

**Metrics.** For evaluation, we used the following standard metrics (Shafer & Vovk, 2008):

(i) **Coverage** is the proportion of test points where the true label is in the prediction set, i.e., $\frac{1}{|\mathcal{D}_{\text{test}}|} \sum_{i \in \mathcal{D}_{\text{test}}} \mathbf{1}[y_i \in \mathcal{C}(\boldsymbol{x}_i)]$.

(ii) **Efficiency** is the average prediction set size, i.e., $\frac{1}{|\mathcal{D}_{\text{test}}|} \sum_{i \in \mathcal{D}_{\text{test}}} |\mathcal{C}(\boldsymbol{x}_i)|$

(iii) **Label (or Class) Stratified Coverage** (Sadinle et al., 2019) is the mean of the coverage for each class, i.e., $\frac{1}{|\mathcal{D}_{\text{test}}|} \sum_{i \in \mathcal{D}_{\text{test}}} \left( \frac{1}{K} \sum_{k=1}^{K} \mathbf{1}[y_i \in \mathcal{C}(\boldsymbol{x}_i), y_i = k] \right)$.

**Dataset Splits and Training.** There are several methods of partitioning $\mathcal{D}$ into $\mathcal{D}_{\text{train}}$, $\mathcal{D}_{\text{valid}}$, $\mathcal{D}_{\text{calib}}$, and $\mathcal{D}_{\text{test}}$. Two methods used in existing works on graph conformal prediction for node classification are:

(i) **Full-Split (FS) Partitioning** (Huang et al., 2023). In this split style, $\mathcal{D}$ is partitioned based on a size constraint. For example, in CFGNN (Huang et al., 2023) the authors split the datasets in their experiments randomly, satisfying a 20%/10%/35%/35% constraint for $\mathcal{D}_{\text{train}}/\mathcal{D}_{\text{valid}}/\mathcal{D}_{\text{calib}}/\mathcal{D}_{\text{test}}$. Note that a large portion (65%) of the full dataset has labels provided (i.e., in the development or calibration set). This style is ideal for methods with numerous trainable (or tunable) parameters, as more calibration data is available for training. We consider the following splits: $(\mathcal{D}_{\text{train}}, \mathcal{D}_{\text{valid}}, \mathcal{D}_{\text{calib}}, \mathcal{D}_{\text{test}})$ $= (0.2, 0.1, 0.35, 0.35), (0.2, 0.2, 0.3, 0.3), (0.3, 0.1, 0.3, 0.3)$, and $(0.3, 0.2, 0.25, 0.25)$.

(ii) **Label-Count (LC) Sample Partitioning** (Zargarbashi et al., 2023). In this split style, the data is partitioned to ensure an equal number of samples of each label is present in $\mathcal{D}_{\text{train}}$, $\mathcal{D}_{\text{valid}}$, and $\mathcal{D}_{\text{calib}}$. The remaining nodes are $\mathcal{D}_{\text{test}}$. This setting is common when only a small proportion of labeled nodes are available (e.g., semi-supervised learning). Intuitively, this setting is ideal for methods that do not have many parameters to train. We explore setting the number of samples per class to 10, 20, 40, and 80. Note that we assign nodes of each class sequentially, so it is feasible in this setup to have some classes with no representative samples in some data subset. Finally, we note that this is a biased sampling scheme which may violate the exchangeability assumption; however, we include this scheme as it is used in the literature (Zargarbashi et al., 2023; 2024) for completeness of our analysis.

If the dataset has **predefined splits** (e.g., Flickr, ogbn-arxiv), we ensure that $\mathcal{D}_{\text{train}}$ and $\mathcal{D}_{\text{valid}}$ come solely from the training and validation splits, while $\mathcal{D}_{\text{calib}} \cup \mathcal{D}_{\text{test}}$ come from the test split.

**3.1. Proposed Baseline using TPS with Adaptability.** From Figure 1, we see that using TPS-Classwise successfully provides label-stratified coverage in both the FS and LC split settings. This additional adaptability comes at a cost to efficiency as seen in Figure 7, a standard tradeoff in conformal prediction. One factor that impacts adaptability is the dataset's label distribution, particularly because of labels with little representation, as seen in Figure 2 for ogbn-arxiv. Because TPS is not adaptive, it has been ignored as a baseline; however, TPS-Classwise's adaptability makes it a viable baseline for future scholarship.

**3.2. Empirical Observations on Randomization for APS.** Applying Theorem 2.3 to randomized APS ($A$) and deterministic APS ($\tilde{A}$), intuitively, as each score in $A$ gets shifted by a small $u\pi$ term to the left, $q_A$ would be lower than $q_{\tilde{A}}$. Thus, the miscoverage levels we would search for in the complementary scores $1 - \alpha_c^A$ would be less than $1 - \alpha_c^{\tilde{A}}$. $1 - \alpha_c^A < 1 - \alpha_c^{\tilde{A}} \implies \alpha_c^A - \alpha_c^{\tilde{A}} > 0$. If the shift is sufficiently large, the randomized prediction set is more efficient than the non-randomized one.

In Figure 3, we show what this looks like for a practical example over the Cora dataset. In Figure 3 (right), the normalized sorted index at which the lower threshold $q_A$ is reached when considering the incorrect classes is lower, i.e., $1 - \alpha_c^A$ is lower, and hence $\alpha_c^A$ is higher. As a part of the proof, we show dependencies on $\frac{1}{(n+1)}$ and $(K-1)$, which indicates the improvements are more pronounced for larger $\mathcal{D}_{\text{calib}}$ and many classes.

Figure 4 provides box plots that compare the efficiency of randomized and non-randomized versions of APS across different datasets. We observe that the randomized version consistently provides a more efficient prediction set for each split type. This effect is most pronounced for datasets with many potential classes in the FS split, which aligns with the intuition from Theorem 2.3 described above and Corollary 2.3.1. Overall, the empirical results show that the effect of randomized APS is more pronounced for larger values of $K$. This

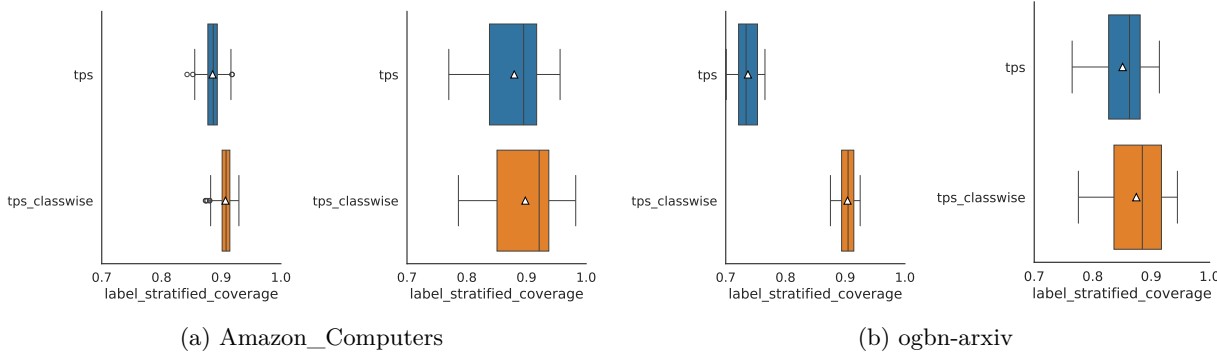

(a) Amazon_Computers                                    (b) ogbn-arxiv

Figure 1: We set the target coverage rate $\alpha = 0.1$. The boxplots present the Label Stratified Coverage for (a) Amazon_Computers and (b) ogbn-arxiv for both the FS split (left) and LC split (right). We want the means (white triangle) to be around $1 - \alpha = 0.9$. For Labeled Stratified Coverage, TPS-Classwise is comparable to or better than TPS.

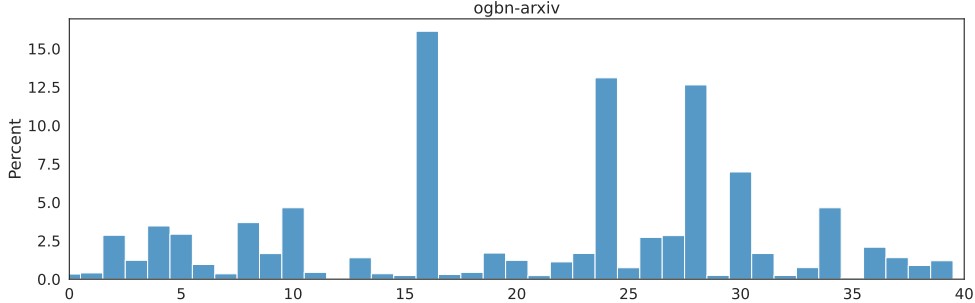

Figure 2: Label Distribution for ogbn-arxiv

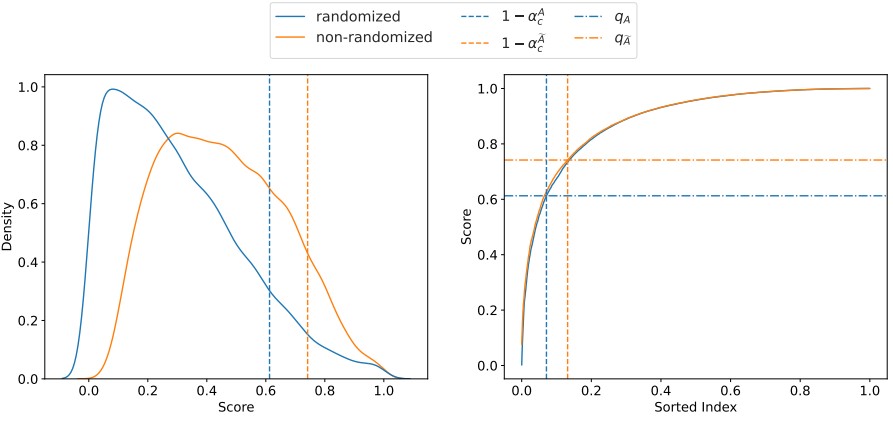

Figure 3: Scores for the Cora dataset using the randomized and non-randomized versions of APS. In the left plot, the vertical lines show the shift in the standard conformal quantiles for $A$ (randomized APS) and $\tilde{A}$ (non-randomized APS) for 0.9 coverage. In the right plot, the vertical lines show the shift in the $1 - \alpha_c$ value for $A$ and $\tilde{A}$ using scores for the incorrect classes. We have $(1 - \alpha_c^{\tilde{A}}) - (1 - \alpha_c^A) \gg \frac{2}{n+1} \iff \alpha_c^A - \alpha_c^{\tilde{A}} \gg \frac{2}{n+1}$ which satisfies the condition for Theorem 2.3. Thus, $A$ is more efficient as seen in the left plot since $q_A < q_{\tilde{A}}$.

observation suggests that APS with randomization should be the default implementation of APS for future scholarship when looking to optimize (prediction) efficiency.

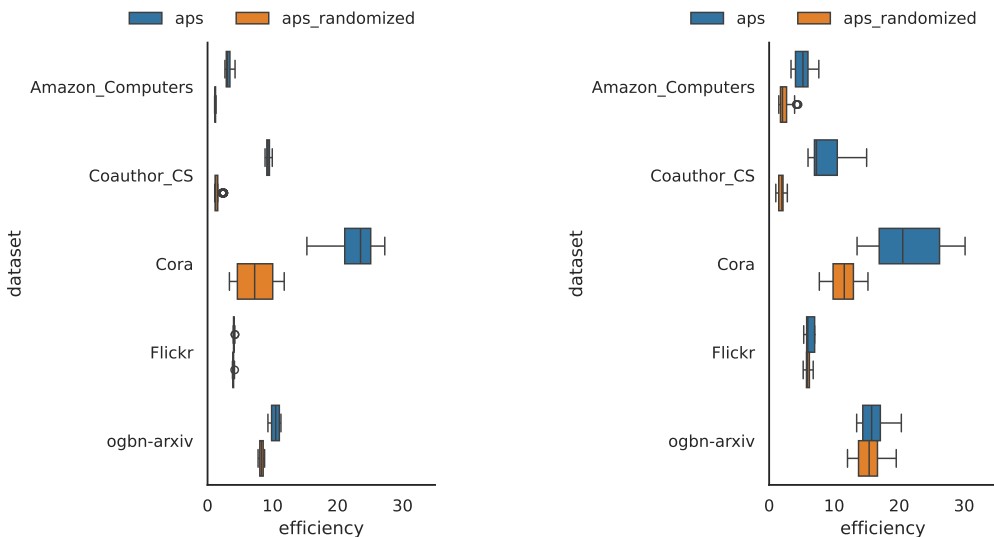

Figure 4: We set the target coverage rate $\alpha = 0.1$. Box plots depicting the efficiencies of APS and Randomized APS across different datasets and multiple runs in both the FS split (left) and LC split (right). Using randomization (the lower box plot for each dataset) consistently improves over the non-randomized version as the efficiencies are distributed around smaller values.

**3.3  Impact of Baseline and Scoring Function in CFGNN Inefficiency Loss.** The choice of non-conformity score in the inefficiency loss during calibration and testing plays a vital role in determining the overall performance of GNN-based conformal prediction. To illustrate, we replicate an experiment by Huang et al. (2023) who use TPS for the inefficiency loss during the calibration stage and non-randomized APS for the testing stage, which gives a significant improvement in efficiency over the baseline (Figure 5 right). However, if randomized APS is used instead (Figure 5 left), we observe that the baseline is competitive across various coverage thresholds. It is worth noting that CFGNN appears robust to this choice, although the gains in efficiency are not as dramatic in the randomized setting. We also note that the confidence bars are narrower in the randomized setting. Similar results were observed on other datasets (see Appendix E).

Based on these insights, we use an improved version of CFGNN, which uses APS with randomization for *both* training and evaluation, labeled as 'cfgnn_aps.' The original implementation is labeled as 'cfgnn_orig'. Our library implementation of CFGNN allows for either TPS or APS to be used for calibration and testing and is extensible to other conformal prediction methods. We extend our results to more datasets and compare against 'aps_randomized' (i.e., the baseline without applying the topology-aware correction via CFGNN). Figure 6 presents the results for both the FS and LC split styles. We observe that 'cfgnn_aps' is as good as 'cfgnn_orig' and can even improve upon 'aps_randomized' in terms of efficiency. In the LC split setting, CFGNN is seen to be quite brittle because there may not be enough data to train a second GNN.

**3.4  Scaling CFGNN for Large Graphs.** The original CFGNN implementation uses full batch training. While this approach has merits, it also poses challenges, particularly when dealing with larger graphs. The evident need for a more scalable solution led us to implement modifications. We implemented a batched version of CFGNN to ensure it can be used for larger graphs (e.g., ogbn-arxiv). To further scale CFGNN, we cache the outputs from the base model to be treated as features for CFGNN training rather than having to sample neighbors for both the base model and CFGNN, significantly speeding up the computation in training and evaluation.

We compare three CFGNN implementations to demonstrate the impact of batching and caching on the runtime. Across all comparisons, we use the FS split, with a 20%/20%/35%/25% data split. We use the best base GNN (w.r.t. accuracy) and best CFGNN (w.r.t. efficiency) architecture from hyperparameter tuning. The baseline implementation follows the setup by Huang et al. (2023), where the CFGNN is trained with

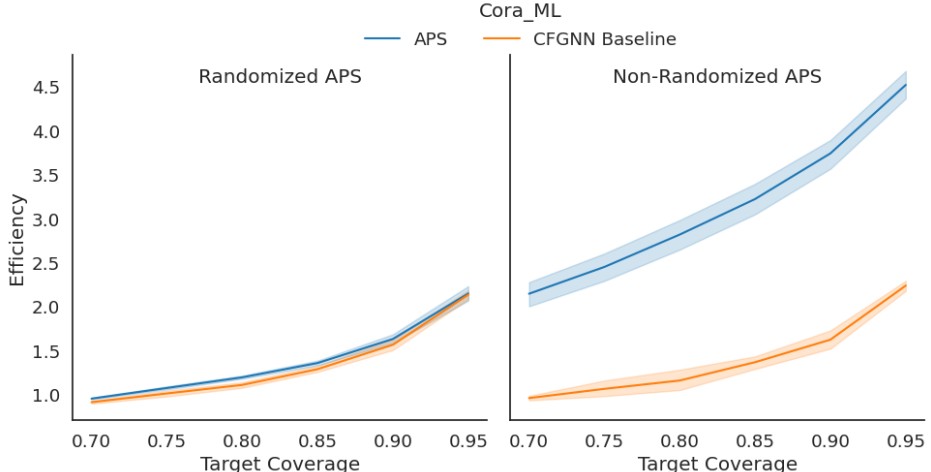

Figure 5: The plot on the right replicates an experiment (Huang et al., 2023) plotting efficiency over various coverage rates for the Cora_ML dataset (a subset of the Cora dataset) for both CFGNN and a baseline model. The plot on the left uses APS with randomization when constructing the final prediction sets. These plots illustrate the benefits of using randomization on baseline performance.

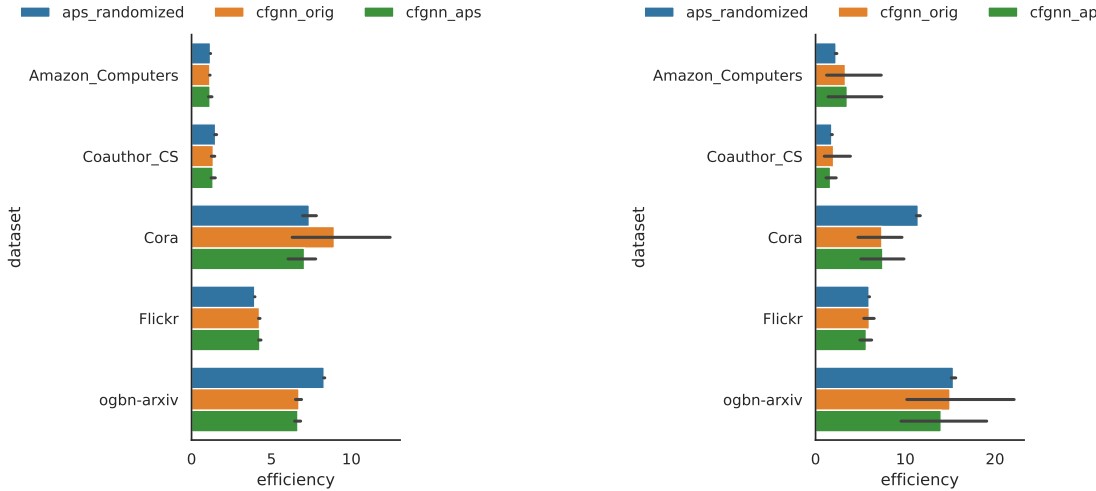

Figure 6: Bar charts denoting efficiency for 'cfgnn_aps', 'cfgnn_orig', and 'aps_randomized' for the FS split (left) and LC split (right) at $\alpha = 0.1$. We see that 'cfgnn_aps' improves or matches efficiency in most cases.

full batch gradient descent for 1000 epochs. Our mini-batch implementation achieves comparable efficiency within 50 epochs without any batch size tuning (we set the batch size to 64 for consistent comparison) as shown in Table 2. Finally, we cache the output probabilities from the base GNN on top of the batched implementation, reducing the runtime further. Table 2 compares the batching and the combined batching + caching improvements. Our implementation can achieve improvements ranging from $3.69\times$ (ogbn-arxiv) to $20.16\times$ (Coauthor_CS) in runtime over the baseline implementation[2].

**3.5. Summary of Key Insights For Graph Conformal Prediction.** In Figure 7, we present an overall comparison of the different methods on the Flickr dataset in terms of efficiency and label-stratified coverage (i.e., adaptability) using the full-split data partitioning style. Similar figures for the other datasets can be

---

[2]We provide config files for the best base GNN and CFGNN architectures in the attached code for every dataset/split type.

Table 2: Impact of different CFGNN implementations between the original, batching, and batching+caching. Used the **best** CFGNN architecture (w.r.t. validation efficiency) for each dataset. We run 5 trials for each setup and report a 95% confidence interval. Amzn_Comp = Amazon_Computers

| method($\rightarrow$) | original | | batching | | batching+caching | |
|---|---|---|---|---|---|---|
| dataset($\downarrow$) | Runtime | Efficiency | Runtime | Efficiency | Runtime | Efficiency |
| Amzn_Comp | $664.98 \pm 10.90$ | $1.29 \pm 0.05$ | $205.29 \pm 3.96$ | $1.15 \pm 0.01$ | $73.85 \pm 1.56$ | $1.14 \pm 0.01$ |
| Cora | $1378.01 \pm 13.35$ | $6.81 \pm 1.07$ | $203.61 \pm 2.52$ | $8.34 \pm 1.12$ | $73.60 \pm 1.53$ | $7.96 \pm 0.59$ |
| Coauthor_CS | $638.56 \pm 6.14$ | $1.10 \pm 0.01$ | $88.49 \pm 1.14$ | $1.15 \pm 0.01$ | $31.67 \pm 0.53$ | $1.14 \pm 0.01$ |
| Flickr | $868.87 \pm 9.98$ | $4.23 \pm 0.07$ | $567.39 \pm 6.50$ | $4.23 \pm 0.04$ | $56.71 \pm 1.30$ | $4.24 \pm 0.03$ |
| ogbn-arxiv | $410.91 \pm 8.29$ | $7.07 \pm 0.05$ | $373.19 \pm 3.64$ | $7.28 \pm 0.06$ | $111.38 \pm 1.95$ | $6.91 \pm 0.01$ |

found in Appendix E. The following sections will analyze some of these methods in more detail, while a discussion on others can be found in Appendix C.

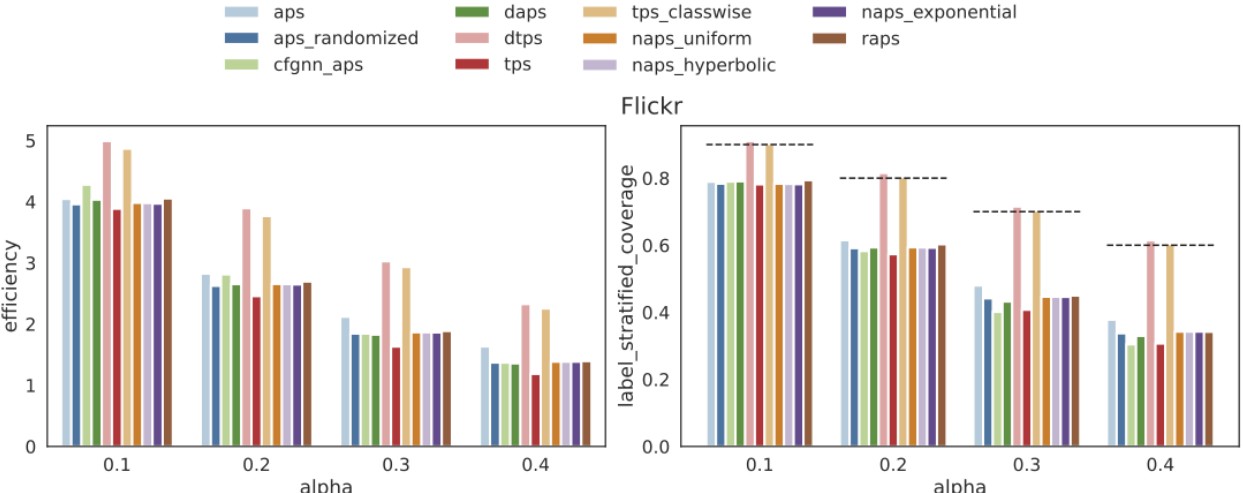

Figure 7: Efficiency (left) and Label Stratified Coverage (right) for all the conformal methods for the Flickr dataset. This uses the FS split style with multiple values for $\alpha$. The dashed black line indicates the desired label-stratified coverage. Other NAPS variants (i.e., exponential, hyperbolic) are discussed in Appendix C.

We analyze several general and graph-specific CP methods proposed in the recent literature for transductive node classification. Based on our theoretical, empirical, and methodological results, we encourage practitioners to be precise and careful when developing and evaluating conformal prediction methods for graphs. We analyze the efficiency of all the methods considered in different datasets. Some key insights we report include:

- **TPS is Efficient.** TPS is consistently the most efficient method for each dataset, regardless of the data split. However, this often comes at a cost to adaptability, as shown in Figure 7 for the Flickr dataset.

- **TPS-Classwise is a viable baseline.** While TPS is provably optimal for efficiency, the classwise version has largely been ignored in the literature as a baseline. Our results demonstrate that TPS-Classwise achieves label-stratified coverage–stemming from its classwise quantile procedure– making it an 'adaptive' method that is a competitive baseline. This adaptability usually comes at a small cost to efficiency, though this is not the case with all datasets (see PubMed in Figure E5).

- **Diffused Classwise TPS.** We also explored an alternative to classwise TPS where we apply the diffusion operator from Zargarbashi et al. (2023) on top of the 'adaptive' TPS-Classwise, discussed earlier. As with

other datasets, DTPS also provides label-stratified coverage for Flickr (see Figure 7). More discussions on DTPS and Diffusion Adaptive Sets (DAPS) (Zargarbashi et al., 2023) can be found in Appendix C.1.

- **APS with randomization is provably more efficient than APS without randomization and should be the default setting for future evaluations involving APS.** From Theorem 2.3, we observe that the randomization term in APS Randomized decreases the incorrect label scores *substantially less* than the conformal quantile, meaning these incorrect labels are less likely to be included when constructing the prediction set, therefore improving efficiency. This result is further pronounced for datasets with many classes because most incorrect labels shift less than the correct label, suggesting that APS Randomized should be a default baseline in the future.

- **NAPS provides a balanced tradeoff (and applies to the non-exchangeable setting).** Using the NAPS (Clarkson, 2023) scoring function achieves a balance between efficiency and label-stratified coverage. Before computing a quantile, NAPS uses neighbor distances to assign weights to non-conformity scores. In the original paper and our implementation, NAPS has a hyperparameter $k$, which is the maximum distance considered when assigning a weight, and everything beyond that is given zero weight. A unique characteristic of NAPS is that it is the only method discussed that provides guarantees for the non-exchangeable setting. More details on NAPS and its variations can be found in Appendix C. Appendix E has similar plots for the other datasets.

- **Randomization in CFGNN.** As noted in Figure 6, randomization within the inefficiency loss function can improve the efficiency of the original CFGNN implementation, particularly for datasets with many classes (e.g., Cora). It is also worth reiterating that the perceived efficiency gains of the original CFGNN are not as significant compared to APS with randomization (see Figure 5).

- **Scaling up CFGNN.** We reimplement CFGNN using batching and caching to scale the method for larger graphs (e.g., ogbn-arxiv). We observe runtime improvements (see Table 2) and, in some cases, batching with caching is the only method that can run CFGNN (e.g., ogbn-products in Table C1a). Moreover, Figure 6 shows that APS Randomized is slightly less efficient than CFGNN without the same computational demands. Thus, scaling makes CFGNN more viable for settings with limited resources.

- **Impact of Label Distribution.** We observe that the dataset's label distribution impacts the difference in adaptability between TPS and TPS-Classwise. In Figure 1, we find an increase in the adaptability of TPS when using LC split instead of FS split for ogbn-arxiv compared to Amazon_Computers. The shift in performance is exacerbated when many classes have little representation (highly imbalanced), as seen in Figure 2. The difference in adaptability is also observed in other non-conformity scores and is particularly pronounced for the ogbn-products dataset (see Figure D1).

- **Incorrect label miscoverages are important.** Conformal prediction is concerned with the true label miscoverage; however, since prediction sets also include incorrect labels, the miscoverage levels of incorrect labels are important to account for when considering a method's efficiency. With Theorem 2.3, we prove that a sufficient condition for when a CP method is more efficient than another method depends on the incorrect label miscoverage. Theorem 2.3 applies to other domains outside the graph space. We also see in Corollary 2.3.1 that the incorrect label miscoverages determine how much the efficiency improves with large calibration sets. This improvement is pronounced for datasets with many classes.

- **The Benefit of RAPS is not as significant for Transductive Node Classification.** In the general CP literature, RAPS has demonstrated improvements to (prediction) efficiency over APS consistently, however, these same improvements aren't as evident for graph datasets as seen in Figure 7 where RAPS does not beat out APS Randomized in terms of (prediction) efficiency. Similar results for other datasets can be found in Appendix E and Zargarbashi et al. (2023).

**Guidelines.** To conclude this analysis, if computational resources are not a limiting factor, CFGNN with the appropriate loss functions (e.g., APS randomized) has demonstrable success in improving efficiency, while being comparable to other methods in terms of adaptability. When deciding which method is more efficient in general, Theorem 2.3 can be applied to the calibration set to determine this prior to deployment. If

adaptability is critical, then methods such as TPS-Classwise and DTPS are both viable and the only methods that provide *a guarantee* while not being as computationally expensive. APS and APS Randomized are good middle-ground methods, and NAPS is the only method that provides guarantees for non-exchangeable settings. Lastly, we provide a Python library[3] which can be used to implement these methods.

## 4    Concluding Remarks

We present a comprehensive benchmarking study of conformal prediction for node classification. We provide novel insights related to design choices that impact efficiency, adaptability, and scalability. Along the way, we offer a new theoretical rationale for the importance of randomization and discuss some novel methodological improvements and directions for future work. One future direction pertains to the space of fairness auditing. Several works have dealt with the auditing fairness of ML models through measuring uncertainty in fairness definitions (Ghosh et al., 2021; Maneriker et al., 2023; Yan & Zhang, 2022), but they rely on the assumption of IID. While conformal prediction works with the notion of *miscoverage*, more relevant notions of error can be considered using the generalized framework of conformal risk control (Angelopoulos et al., 2024).

### Acknowledgments

The authors acknowledge support from National Science Foundation (NSF) grant #2112471 (AI-EDGE) and a grant from Cisco Research (US202581249). Any opinions and findings are those of the author(s) and do not necessarily reflect the views of the granting agencies. The authors also thank the anonymous reviewers for their constructive feedback on this work.

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

## A  Optimal $\tau$ for APS

The most popular baseline in the graph conformal prediction literature is Adaptive Prediction Sets (APS). (Romano et al., 2020) introduces APS by defining an optimal prediction set construction mechanism under oracle probability. Suppose we estimate a prediction function $\hat{f}$ that correctly models the oracle probability $\Pr[Y = y | X_{test} = \boldsymbol{x}] = \pi_y(\boldsymbol{x})$ for each $y \in \mathcal{Y} = \{1, \ldots, K\}$ Let $\pi_{(1)}(\boldsymbol{x}), \ldots, \pi_{(K)}(\boldsymbol{x})$ be the sorted probabilities in descending order. For any $\tau \in [0, 1]$, define the generalized conditional quantile funciton at $\tau$ as

$$L(x; \pi, \tau) = \min \left\{ k \in \{1, \ldots, K\}, \sum_{j=1}^{k} \pi_{(j)}(\boldsymbol{x}) \geq \tau \right\} \tag{5}$$

The corresponding prediction set, $\mathcal{C}_\alpha^{\mathrm{or}}(\boldsymbol{x})$, is constructed as

$$\mathcal{C}_\alpha^{\mathrm{or}+}(\boldsymbol{x}) = \{y \in \mathcal{Y} : \pi_y(\boldsymbol{x}) \geq \pi_{(L(\boldsymbol{x};\pi,1-\alpha))}(\boldsymbol{x})\}$$

where or indicates the usage of the oracle probability. Further, they define tighter prediction sets in a randomized fashion using an additional uniform random variable $u \sim \mathrm{Uniform}(0, 1)$ as a parameter to construct a generalized inverse. This idea draws upon the idea of uniformly most powerful tests in the Neyman-Pearson lemma for level-$\alpha$ sets (Neyman & Pearson, 1933). Define

$$S(\boldsymbol{x}, u; \pi, \tau) = \begin{cases} \{y \in \mathcal{Y} : \pi_y(\boldsymbol{x}) > \pi_{(L(\boldsymbol{x};\pi,\tau))}(\boldsymbol{x})\} & u < V(\boldsymbol{x}; \pi, \tau) \\ \{y \in \mathcal{Y} : \pi_y(\boldsymbol{x}) \geq \pi_{(L(\boldsymbol{x};\pi,\tau))}(\boldsymbol{x})\} & \text{otherwise} \end{cases} \tag{6}$$

i.e., the class at the $L(\boldsymbol{x}; \pi, \tau)$ rank is included in the prediction set with probability $1 - V(\boldsymbol{x}; \pi, \tau)$, where

$$V(\boldsymbol{x}; \pi, \tau) = \frac{1}{\pi_{(L(\boldsymbol{x};\pi,\tau))}(x)} \left\{ \left[ \sum_{j=1}^{L(\boldsymbol{x};\pi,\tau)} \pi_{(j)}(x) \right] - \tau \right\}$$

The corresponding randomized prediction sets are $\mathcal{C}_\alpha^{\mathrm{or}}(\boldsymbol{x}) = S(\boldsymbol{x}, U; \pi, 1 - \alpha)$, $U \sim U(0, 1)$ Note, the coverage guarantees provided in conformal prediction hold only in expectation over the randomness in $(\boldsymbol{x}_i, y_i), i = 1, \ldots, n+1$. The randomized prediction sets continue to provide the guarantee with additional randomness over $u_i$. To make this work for a non-oracle probability $\hat{\pi}(\boldsymbol{x})$, they define a non-conformity score $A$

$$A(\boldsymbol{x}, y, u; \hat{\pi}) = \min\{\tau \in [0, 1] : y \in S(\boldsymbol{x}, u; \hat{\pi}, \tau)\} \tag{7}$$

Assume that $\hat{\pi}$ are all distinct, for ease of defining rank. Suppose the rank of the true class amongst the sorted $\hat{\pi}$ be $r_y$, i.e., $\sum_{i=1}^{K} \mathbf{1}[\hat{\pi}_i(\boldsymbol{x}) \geq \hat{\pi}_y] = r_y$ Solving for $\tau$ as a function of $\hat{\pi}$ (see Appendix A, for proof),

$$A(\boldsymbol{x}, y, u; \hat{\pi}) = \left[ \sum_{i=1}^{r_y} \hat{\pi}_{(i)}(\boldsymbol{x}) \right] - u\hat{\pi}_y \tag{8}$$

Instead, if a deterministic set is used to define the conformal score instead (i.e., the randomized set construction is not carried out), then we could add the probabilities until the true class is included:

$$\tilde{A}(\boldsymbol{x}, y; \hat{\pi}) = \left[ \sum_{i=1}^{r_y} \hat{\pi}_{(i)}(\boldsymbol{x}) \right] \tag{9}$$

This version of APS still provides the same conditional coverage guarantees and has a simpler exposition, as the prediction sets are constructed by greedily including the classes until the true label is included. Thus, this version is implemented in the popular monographs on conformal prediction (Angelopoulos & Bates, 2023).

However, the lack of randomization may sacrifice efficiency. This modification of the score function affects both the quantile threshold computation during the calibration phase and the prediction set during the test phase. We will now show the conditions that impact the efficiency more formally.

For simplicity, assume that the probabilities are distinct.

From the definition of $A$ equation 7

$$A(\boldsymbol{x}, y, u; \hat{\pi}) = \min\{\tau \in [0,1] : y \in S(\boldsymbol{x}, u; \hat{\pi}, \tau)\}$$

Define

$$\Sigma_{\hat{\pi}}(\boldsymbol{x}, m) = \sum_{i=1}^{m} \hat{\pi}_{(i)}(\boldsymbol{x})$$

From the definition of $S(\boldsymbol{x}, u; \hat{\pi}, \tau)$ from equation 6, conisder the following cases:

**Case 1:** $\tau = \Sigma_{\hat{\pi}}(\boldsymbol{x}, r_y)$, then $L(x; \hat{\pi}, \tau) = y$ and thus, $V(\boldsymbol{x}; \pi, \tau) = 0$. Thus $\Pr[u > V(\boldsymbol{x}; \pi, \tau)] = 1$ and hence, $P[y \in S(\boldsymbol{x}, u; \hat{\pi}, \tau)] = 1$.

**Case 2:** $\tau = \Sigma_{\hat{\pi}}(\boldsymbol{x}, r_y - 1)$, then $y \notin S(\boldsymbol{x}, u, \hat{\pi}, \tau)$ in either case, since only classes with $\hat{\pi}_i(\boldsymbol{x}) > \hat{\pi}_y(\boldsymbol{x})$ could be included.

**Case 3:** $\tau = \Sigma_{\hat{\pi}}(\boldsymbol{x}, r_y) - \varepsilon\hat{\pi}_y$. Then we have $L(x; \hat{\pi}, \tau) = y$ again, and

$$
\begin{aligned}
V(\boldsymbol{x}; \pi, \tau) &= \frac{1}{\hat{\pi}_y(\boldsymbol{x})} \left\{ \left[ \sum_{j=1}^{r_y} \hat{\pi}_{(j)}(\boldsymbol{x}) \right] - \tau \right\} \\
&= \frac{1}{\hat{\pi}_y(\boldsymbol{x})} \left\{ \left[ \sum_{j=1}^{r_y} \hat{\pi}_{(j)}(\boldsymbol{x}) \right] - (\Sigma_{\hat{\pi}}(\boldsymbol{x}, r_y) - \varepsilon\hat{\pi}_y) \right\} \\
&= \varepsilon
\end{aligned}
$$

For $y$ to be included in $S(\boldsymbol{x}, u; \hat{\pi}, \tau)$, we would require that $u \geq V(\boldsymbol{x}; \pi, \tau)$, i.e., $u \geq \varepsilon$. We want the minimal $\tau$ (or the maximal $\varepsilon$). Thus, $\tau = \Sigma_{\hat{\pi}}(\boldsymbol{x}, r_y) - u\hat{\pi}_y$ is the required solution.

### A.1 Non-randomized set

The inclusion criterion for the score given the threshold $\tau$ is $\tilde{A}(\boldsymbol{x}, y; \hat{pi}) \leq \tau$

To include the correct label $y_i$ while minimizing the chosen threshold $\tau$, we would require $\tau = \sum_{j=1}^{r_{y_i}} \hat{\pi}_{(j)}(\boldsymbol{x})$

# B Proofs

## B.1 Proof of Theorem 2.3

For self-containment, we will restate some of the notation used in proving Theorem 2.3. Let $A(\boldsymbol{x}, y)$ be any non-conformity score function and

$$\hat{q}_A = \text{Quantile}\left(\frac{\lceil(n+1)(1-\alpha)\rceil}{n}; \{A(\boldsymbol{x}_i, y_i)\}_{i=1}^n\right)$$

Define $A_i(y) := A(\boldsymbol{x}_i, y)$ where $y \in \mathcal{Y}$ and $C_A^i = C_A(\boldsymbol{x}_i)$ for brevity. Let $\mathcal{Y}_i' = \mathcal{Y} \setminus \{y_i\}$ be the set of incorrect labels and $y_i' \in \mathcal{Y}_i'$ be any incorrect class label for each $\boldsymbol{x}_i$. We can consider the exchangeable sequence $\{A(\boldsymbol{x}_i, y_i^R)\}_{i=1}^{n+1}$, where $y_i^R$ denotes a randomly sampled from $\mathcal{Y}_i'$, and define $\alpha_c^A \in [0, 1]$ such that:

$$\hat{q}_A = \text{Quantile}\left(\frac{\lceil(n+1)(1-\alpha_c^A)\rceil}{n}; \{A(\boldsymbol{x}_i, y_i^R)\}_{i=1}^n\right). \tag{10}$$

Particularly, if we set $\lambda = \hat{q}_A$ in the following lemma, then we can compute $\alpha_c^A = \frac{\sum_{i=1}^n \mathbf{1}[A(\boldsymbol{x}_i, y_i^R) > \hat{q}_A] + 1}{n+1}$ to satisfy the guarantee given in Equation 4.

**Lemma B.1** (Vadlamani et al. (2025)). *For $\lambda \in [0, 1]$ and $n = |\mathcal{D}_{\text{calib}}|$, let $C_\lambda(\boldsymbol{x}) = \{y \in \mathcal{Y} : s(\boldsymbol{x}, y) \leq \lambda\}$. Then,*

$$\frac{\sum_{i=1}^n \mathbf{1}[s(\boldsymbol{x}_i, y_i) > \lambda]}{n+1} \leq \Pr[y_{n+1} \notin C_\lambda(\boldsymbol{x}_{n+1})] \leq \frac{\sum_{i=1}^n \mathbf{1}[s(\boldsymbol{x}_i, y_i) > \lambda] + 1}{n+1}. \tag{11}$$

**On the exchangeability of $\{A(\boldsymbol{x}_i, y_i)\}_{i=1}^{n+1}$** To understand why $\{A(\boldsymbol{x}_i, y_i^R)\}_{i=1}^{n+1}$ is exchangeable we first consider Lemma B.2:

**Lemma B.2.** *The random variable $y_i^R \sim U(\mathcal{Y}_i')$.*

*Proof.* To prove the lemma, we will show that the PMF that $y_{i,(u_i)}'$ follows is identical to $U(\mathcal{Y}_i')$. Let $y_i'$ be an arbitrary label in $\mathcal{Y}_i'$. First observe for $y_i^R \sim U(\mathcal{Y}_i')$ that $\Pr[y_i^R = y_i'] = \frac{1}{K-1}$ because it is uniformly sampled. Now consider via the law of total probability,

$$
\begin{aligned}
\Pr[y_{i,(u_i)}' = y_i'] &= \sum_{l=1}^{K-1} \Pr[y_{i,(u_i)}' = y_i' | u_i = l] \cdot \Pr[u_i = l] \\
&= \sum_{l=1}^{K-1} \Pr[y_{i,(l)}' = y_i'] \cdot \Pr[u_i = l] \\
&= \sum_{l=1}^{K-1} \Pr[y_{i,(l)}' = y_i'] \frac{1}{K-1} \\
&= \frac{1}{K-1} \underbrace{\sum_{l=1}^{K-1} \Pr[y_{i,(l)}' = y_i']}_{\text{Sum of a PMF}} \\
&= \frac{1}{K-1} \cdot 1 = \frac{1}{K-1}
\end{aligned}
\tag{12}
$$

Thus, the PMFs are identical and $y_i^R \sim U(\mathcal{Y}_i')$. □

Recall $(\boldsymbol{x}_i, y_i)_{i=1}^{n+1}$ are assumed to be exchangeable, thus $(\boldsymbol{x}_i, \mathcal{Y}'_i)_{i=1}^{n+1}$ must also be exchangeable since $\mathcal{Y}'_i$ is the complement of $\{y_i\}$. Lastly, consider an IID sequence $u_i \sim U(\{1, \ldots, K-1\})$ for $i = 1, \ldots, n+1$. Since $\{u_i\}_{i=1}^{n+1}$ is IID, we can adjoin it to the covariates and produce an exchangeable sequence of triples $(\boldsymbol{x}_i, \mathcal{Y}'_i, u_i)_{i=1}^{n+1}$.

From this exchangeable sequence, we can compute the non-conformity scores, and $(\{A(\boldsymbol{x}_i, y'_i) \mid y'_i \in \mathcal{Y}'_i\}, u_i)_{i=1}^{n+1}$. Lastly, to get a pointwise representative for each set, let $y_i^R = y'_{i(u_i)}$ such that $A(\boldsymbol{x}_i, y'_{i(1)}) \leq A(\boldsymbol{x}_i, y'_{i(2)}) \leq \cdots \leq A(\boldsymbol{x}_i, y'_{i(K-1)})$ – assuming there is a suitably random tie-breaking method for identical scores. Then, $\{A(\boldsymbol{x}_i, y_i^R)\}_{i=1}^{n+1}$ is an exchangeable sequence. In practice, the process of ordering the $\mathcal{Y}'_i$ labels is not necessary. However, it concretely demonstrates the procedure to form $\{A(\boldsymbol{x}_i, y_i^R)\}_{i=1}^{n+1}$ ensures the sequence is exchangeable where $y_i^R \sim U(\mathcal{Y}'_i)$ using Lemma B.2.

Now, we will prove Theorem 2.3.

**Theorem 2.3.** *If $\alpha_c^A - \alpha_c^{\tilde{A}} \geq \frac{2}{(n+1)}$ then score function $A$ produces a more efficient prediction set than $\tilde{A}$. Formally, $\mathbb{E}\left[|\mathcal{C}_{\tilde{A}}(\boldsymbol{x}_{n+1})| - |\mathcal{C}_A(\boldsymbol{x}_{n+1})|\right] \geq 0$*

*Proof.* Let $\mathcal{C}_A^i = \mathcal{C}_A(\boldsymbol{x}_i)$ for brevity. First consider, $\mathbb{E}\left[|\mathcal{C}_A^{n+1}|\right]$,

$$
\begin{aligned}
\mathbb{E}\left[|\mathcal{C}_A^{n+1}|\right] &= \mathbb{E}\left[\sum_{y \in \mathcal{Y}} \mathbf{1}[y \in \mathcal{C}_A^{n+1}]\right] \\
&= \mathbb{E}\left[\mathbf{1}[y_{n+1} \in \mathcal{C}_A^{n+1}]\right] + \mathbb{E}\left[\sum_{y'_{n+1} \in \mathcal{Y}'_{n+1}} \mathbf{1}[y'_{n+1} \in \mathcal{C}_A^{n+1}]\right] \\
&= \mathbb{E}\left[\mathbf{1}[y_{n+1} \in \mathcal{C}_A^{n+1}]\right] + \sum_{y'_{n+1} \in \mathcal{Y}'_{n+1}} \mathbb{E}\left[\mathbf{1}[y'_{n+1} \in \mathcal{C}_A^{n+1}]\right] \\
&= \Pr[y_{n+1} \in \mathcal{C}_A^{n+1}] + \sum_{y'_{n+1} \in \mathcal{Y}'_{n+1}} \Pr[y'_{n+1} \in \mathcal{C}_A^{n+1}]
\end{aligned}
\tag{13}
$$

Now using Lemma B.2 we have that $y_i^R \sim U(\mathcal{Y}'_i)$. Then, using the law of total probability and the fact that a label $y$ is in $\mathcal{C}_A^{n+1}$ iff $A(\boldsymbol{x}_{n+1}, y) \leq \hat{q}_A$,

$$
\begin{aligned}
\Pr[A(\boldsymbol{x}_{n+1}, y_{n+1}^R) \leq \hat{q}_A] &= \sum_{y'_{n+1} \in \mathcal{Y}'_{n+1}} \Pr[A(\boldsymbol{x}_{n+1}, y_{n+1}^R) \leq \hat{q}_A | y_{n+1}^R = y'_{n+1}] \cdot \Pr[y_{n+1}^R = y'_{n+1}] \\
&= \sum_{y'_{n+1} \in \mathcal{Y}'_{n+1}} \Pr[A(\boldsymbol{x}_{n+1}, y'_{n+1}) \leq \hat{q}_A] \cdot \Pr[y_{n+1}^R = y'_{n+1}] \\
&= \sum_{y'_{n+1} \in \mathcal{Y}'_{n+1}} \Pr[y'_{n+1} \in \mathcal{C}_A^{n+1}] \cdot \Pr[y_{n+1}^R = y'_{n+1}] \\
&= \sum_{y'_{n+1} \in \mathcal{Y}'_{n+1}} \Pr[y'_{n+1} \in \mathcal{C}_A^{n+1}] \cdot \frac{1}{K-1} \\
\Longleftrightarrow (K-1)\Pr[y_{n+1}^R \in \mathcal{C}_A^{n+1}] &= \sum_{y'_{n+1} \in \mathcal{Y}'_{n+1}} \Pr[y'_{n+1} \in \mathcal{C}_A^{n+1}]
\end{aligned}
\tag{14}
$$

Then substituting Equation 14 into Equation 13 we arrive at,

$$
\mathbb{E}\left[|\mathcal{C}_A^{n+1}|\right] = \Pr[y_{n+1} \in \mathcal{C}_A^{n+1}] + (K-1)\Pr[y_{n+1}^R \in \mathcal{C}_A^{n+1}]
\tag{15}
$$

Then we can compute an upper bound and lower bound for $\mathbb{E}\left[\left|\mathcal{C}_A^{n+1}\right|\right]$,

$$\mathbb{E}\left[\left|\mathcal{C}_A^{n+1}\right|\right] \leq 1 - \alpha + \frac{1}{n+1} + (K-1)\left(1 - \alpha_c^A + \frac{1}{n+1}\right)$$

$$= 1 - \alpha + (K-1)\left(1 - \alpha_c^A\right) + \frac{K}{n+1}$$

and

$$\mathbb{E}\left[\left|\mathcal{C}_A^{n+1}\right|\right] \geq 1 - \alpha + (K-1)\left(1 - \alpha_c^A\right) \tag{16}$$

similar bounds can be derived for $\mathbb{E}\left[\left|\mathcal{C}_{\tilde{A}}^{n+1}\right|\right]$. Thus,

$$\mathbb{E}\left[\left|\mathcal{C}_{\tilde{A}}^{n+1}\right| - \left|\mathcal{C}_A^{n+1}\right|\right] \geq \underbrace{\left(1 - \alpha + (K-1)\left(1 - \alpha_c^{\tilde{A}}\right)\right)}_{\text{lower bound for } \mathbb{E}\left[\left|\mathcal{C}_{\tilde{A}}^{n+1}\right|\right]} - \underbrace{\left(1 - \alpha + (K-1)\left(1 - \alpha_c^A\right) + \frac{K}{n+1}\right)}_{\text{upper bound for } \mathbb{E}\left[\left|\mathcal{C}_A^{n+1}\right|\right]}$$

$$= (K-1)\left(\alpha_c^A - \alpha_c^{\tilde{A}}\right) - \frac{K}{n+1}$$

$$= (K-1)\left(\alpha_c^A - \alpha_c^{\tilde{A}} - \frac{K}{(K-1)n+1}\right)$$

$$\geq (K-1)\left(\alpha_c^A - \alpha_c^{\tilde{A}} - \frac{2}{n+1}\right) \geq 0 \tag{17}$$

with the last inequality holding because $\alpha_c^A - \alpha_c^{\tilde{A}} \geq \frac{2}{n+1}$. This completes the proof for the general case. $\square$

### B.2 Proof of Corollary 2.3.1

**Corollary 2.3.1.** *As $n \to \infty$, $\mathbb{E}[|\mathcal{C}_{\tilde{A}}(\boldsymbol{x}_{n+1})| - |\mathcal{C}_A(\boldsymbol{x}_{n+1})|] = (K-1)\left(\alpha_c^A - \alpha_c^{\tilde{A}}\right)$*

*Proof.* Observe using Equation 17, $\mathbb{E}\left[|\mathcal{C}_{\tilde{A}}^{n+1}| - |\mathcal{C}_A^{n+1}|\right] \geq (K-1)\left(\alpha_c^A - \alpha_c^{\tilde{A}} - \frac{2}{n+1}\right)$. Similarly by swapping $A$ and $\tilde{A}$:

$$\mathbb{E}\left[|\mathcal{C}_A^{n+1}| - |\mathcal{C}_{\tilde{A}}^{n+1}|\right] \geq (K-1)\left(\alpha_c^{\tilde{A}} - \alpha_c^A - \frac{2}{n+1}\right)$$

$$\iff \mathbb{E}\left[|\mathcal{C}_{\tilde{A}}^{n+1}| - |\mathcal{C}_A^{n+1}|\right] \leq (K-1)\left(\alpha_c^A - \alpha_c^{\tilde{A}} + \frac{2}{n+1}\right)$$

Then observe as $n \to \infty$ both the lower and upper bound of $\mathbb{E}\left[|\mathcal{C}_{\tilde{A}}^{n+1}| - |\mathcal{C}_A^{n+1}|\right]$ converge to $(K-1)\left(\alpha_c^A - \alpha_c^{\tilde{A}}\right)$. Thus $\mathbb{E}\left[|\mathcal{C}_{\tilde{A}}^{n+1}| - |\mathcal{C}_A^{n+1}|\right] = (K-1)\left(\alpha_c^A - \alpha_c^{\tilde{A}}\right)$. $\qquad\square$

# C Method Details and Innovations

## C.1 Diffusion Adaptive Prediction Sets (DAPS)

The Diffusion Adaptive Prediction Sets (DAPS) approach for conformal node classification on graphs was introduced by (Zargarbashi et al., 2023). The intuition behind DAPS follows the prevalence of homophily graphs, which suggests that non-conformity scores for two connected nodes should be related. DAPS uses a diffusion step to capture this relationship and uses the non-conformity scores modified by diffusion to generate the prediction sets. Formally, suppose $s(v, y)$ is a point wise non-conformity score for a node $v$ and label $y$ (e.g., TPS or APS)

$$\hat{s}(v, y) = (1 - \lambda)s(v, y) + \frac{\lambda}{|\mathcal{N}_v|} \sum_{u \in \mathcal{N}_v} s(u, y)$$

where $\mathcal{N}_v$ is the 1-hop neighborhood of $v$ and $\lambda \in [0, 1]$ is a hyperparameter controlling the diffusion.

Zargarbashi et al. (2023) uses the APS score as the point-wise score in the diffusion process, as it is adaptive and uniformly distributed in $[0, 1]$ under oracle probability. However, as noted earlier, using classwise thresholds provides a mechanism to produce adaptive scores from TPS. Thus, we create DTPS, a variation of DAPS using TPS scores as the point-wise scores in the diffusion process.

We compare our proposed method of using diffusion on top of TPS-Classwise (DTPS) against DAPS, which was proposed by Zargarbashi et al. (2023). From Figure C1 (left), we see that DTPS can be competitive with DAPS in efficiency while providing better label-stratified coverage. However, for some of the larger datasets (Cora, Flickr, ogbn-arxiv, ogbn-products), DTPS suffers from poorer efficiency compared to DAPS. This can be partially explained by the worse performance of the pre-diffusion TPS-classwise (Figure E2), which is forced to sacrifice efficiency on these datasets to achieve label-stratified coverage (Figure E1).

However, when we control the number of samples per class with LC splits (Figure C1 right), we see that DTPS label stratified coverage deteriorates significantly compared to DAPS. Based on these results, we can conclude that DTPS is not a universally better method than DAPS, and its performance is sensitive to the calibration set size and the number of classes. It may be a viable candidate over DAPS in scenarios when there is a sufficiently large calibration set, which is when TPS-classwise has competitive efficiency to TPS.

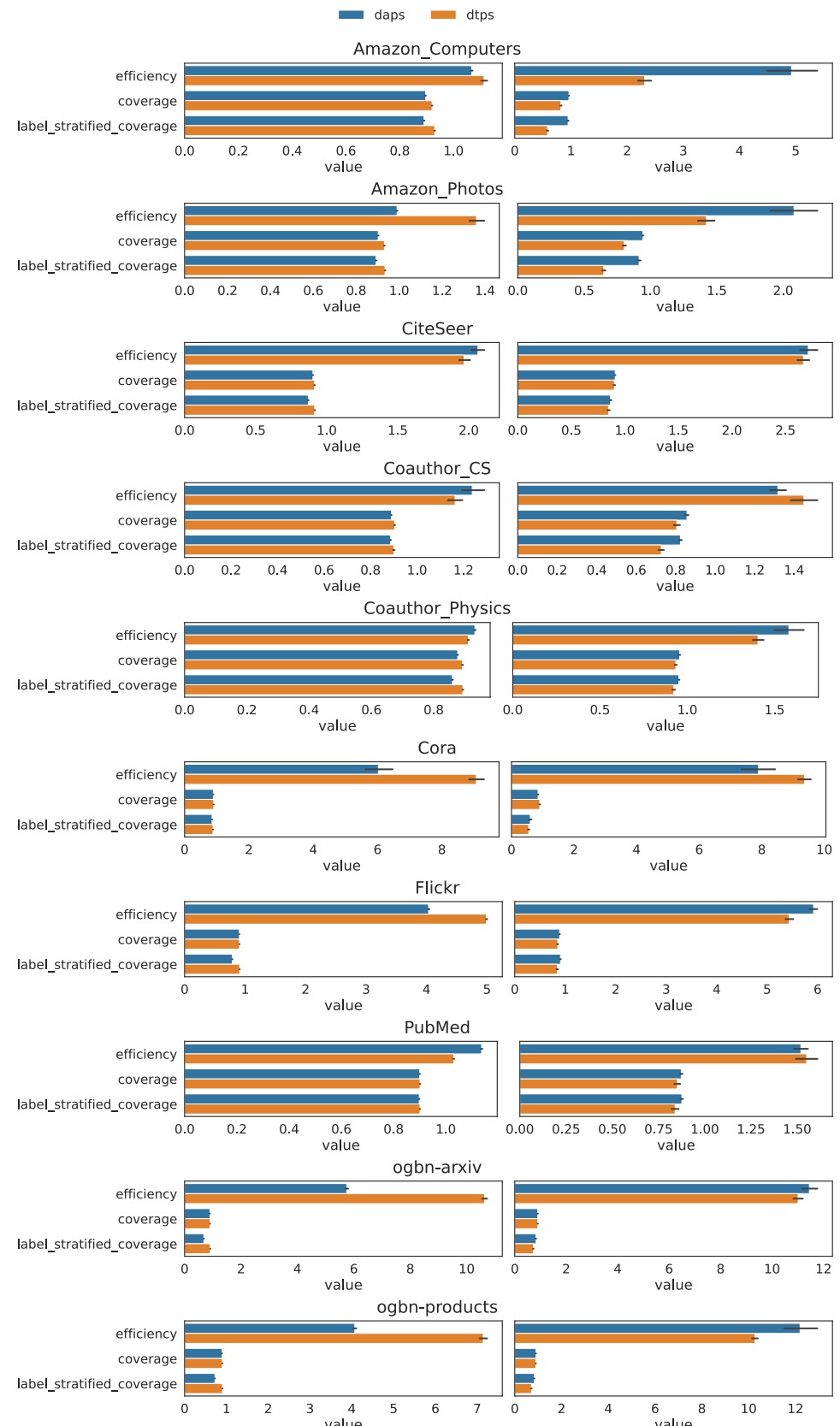

Figure C1: Bar charts denoting different metrics associated with DAPS and DTPS across all datasets for FS split (left) and LC split (right) at $\alpha = 0.1$.

## C.2 Notes on Transductive NAPS

Neighborhood Adaptive Prediction Sets (NAPS) constructs prediction sets under relaxed exchangeability (or non-exchangeability) assumptions (Barber et al., 2023) and was initially implemented for the inductive setting (Clarkson, 2023). However, NAPS can also be used in the transductive setting (Zargarbashi et al., 2023). To compute scores for $\mathcal{D}_{\text{calib}}$ nodes, NAPS uses APS. Using these scores, Equation 18 is used to compute a *weighted quantile* for the score threshold to be used when constructing the prediction sets. The weighted quantile is defined by placing a point mass, $\delta_{s_i}$, for each calibration point's score, $s_i$, as well as a point mass at $\delta_{+\infty}$ to represent the test point, $v_{n+1}$'s, score. The point mass at $\delta_{+\infty}$ is needed because the score for $v_{n+1}$ is unknown, and potentially unbounded, due to non-exchangeability.

$$\hat{q}_{n+1}^{\text{NAPS}} = \text{Quantile}\left(1 - \alpha, \left[\sum_{i \in \mathcal{D}_{\text{calib}}} \tilde{w}_i \cdot \delta_{s_i}\right] + \tilde{w}_{n+1} \cdot \delta_{+\infty}\right) \tag{18}$$

For NAPS to produce viable prediction sets, the weights, $w_i \in [0, 1]$, for the calibration nodes must be chosen in a data-independent fashion, i.e., they cannot leverage the associated node features (Barber et al., 2023). NAPS leverages the graph structure to assign these weights, assigning non-zero weights to nodes within a $k$-hop neighborhood, $\mathcal{N}_{n+1}^k$, of a test node $v_{n+1}$. For nodes that are in $\mathcal{N}_{n+1}^k$, let $d_i$ be the distance from the node to $v_{n+1}$ to $v_i \in \mathcal{V}_{\text{calib}}$. The three implemented weight functions are *uniform*: $w_u(d_i) = \mathbf{1}[d_i \leq k]$, *hyperbolic*: $w_h(d_i) = \frac{1}{d_i}\mathbf{1}[d_i \leq k]$, and *exponential*: $w_e(d_i) = 2^{-d_i}\mathbf{1}[d_i \leq k]$. The weights are then normalized, $\tilde{w}_i$, such that $\sum_{i \in \mathcal{D}_{\text{calib}}} \tilde{w}_i + \tilde{w}_{n+1} = 1$ (Barber et al., 2023).

**NAPS Implementation** NAPS is computationally expensive in terms of time and memory since the $k$-hop intersection is computed for each test node. To allow for scalability, our implementation of NAPS, shown in Algorithms 1 and 2, uses batching to ensure sufficient memory is available and uses sparse-tensor multiplication to reduce memory and time costs.

To ensure scalability for large graphs, all the computations until the quantile computation step were done via sparse tensors. Algorithm 2 illustrates how the distance to each calibration node in the $k$-hop neighborhood can be computed via sparse tensor primitives.

---

**Algorithm 1** NAPS Quantile Implementation

1: **procedure** NAPS_QUANTILE($w, k, \mathcal{D}_{\text{calib}}, \mathcal{D}_{\text{test}}, \mathcal{D}, \mathcal{S}_{\text{calib}}, b, \alpha$)
2:     $\{\mathcal{B}_1, \mathcal{B}_2, \ldots, \mathcal{B}_b\} \leftarrow \text{SPLIT}(\mathcal{D}_{\text{test}}, b)$         ▷ Split test nodes into b batches
3:     $q \leftarrow \text{ZEROS}(\mathcal{D}_{\text{test}}, 1)$         ▷ $q \in \mathbb{R}^{|\mathcal{D}_{\text{test}}| \times 1}$
4:     **for** $\mathcal{B}_n \in \{\mathcal{B}_1, \mathcal{B}_2, \ldots, \mathcal{B}_b\}$ **do**
5:         $\text{k\_hop} \leftarrow \text{SPARSE\_K\_HOP}(k, \mathcal{B}_n, \mathcal{D}_{\text{calib}}, \mathcal{D})$     ▷ $\text{k\_hop} \in \mathbb{R}^{|\mathcal{B}_n| \times |\mathcal{D}_{\text{calib}}|}$
6:         $\text{weights} \leftarrow \text{COMPUTE\_WEIGHTS}(w, \text{k\_hop})$     ▷ $\text{weights} \in \mathbb{R}^{|\mathcal{B}_n| \times |\mathcal{D}_{\text{calib}}|}$
7:         $q[\mathcal{B}_n] \leftarrow \text{COMPUTE\_QUANTILE}(1 - \alpha, \text{weights}, \mathcal{S}_{\text{calib}})$
8:     **end for**
9:     **return** $q$         ▷ Return the quantiles for each test node
10: **end procedure**

---

---

**Algorithm 2** Sparse K Hop Neighborhood Implementation

---

1: **procedure** SPARSE_K_HOP$(k, \mathcal{B}, \mathcal{D}_{\text{calib}}, \mathcal{D})$
2:     $A \leftarrow$ GET_ADJACENCY$(\mathcal{D})$               ▷ Adjacency of $\mathcal{D}$, $A \in \mathbb{R}^{|\mathcal{D}| \times |\mathcal{D}|}$
3:     path_n $\leftarrow A[\mathcal{B}, :]$               ▷ path_n $\in \mathbb{R}^{|\mathcal{B}| \times |\mathcal{D}|}$
4:     k_hop $\leftarrow$ path_n$[:, \mathcal{D}_{\text{calib}}]$               ▷ k_hop $\in \mathbb{R}^{|\mathcal{B}| \times |\mathcal{D}_{\text{calib}}|}$
5:     **for** $n \in \{2, 3, \ldots, k\}$ **do**
6:         path_n $\leftarrow$ (path_n)$A$
7:         neg_if_n $\leftarrow$ k_hop $-$ SGN(path_n$[:, \mathcal{D}_{\text{calib}}]$)          ▷ negative value $\implies$ n hops away
8:         in_n_hop $\leftarrow$ (neg_if_n $< 0$) $\times n$          ▷ Nodes that are a min distance of n
9:         k_hop $\leftarrow$ k_hop $+$ in_n_hop
10:     **end for**
11:     **return** k_hop          ▷ $\forall_{i,j}$ **If** $\text{dist}(i, j) \leq k$ **then** k_hop$[i, j] = \text{dist}(i, j)$, **else** k_hop$[i, j] = 0$
12: **end procedure**

---

**Parameter Analysis:** Apart from the particular weighting function, the main parameter in the NAPS algorithm is the number of hops to consider, $k$. Figure C2 shows the trend between efficiency and coverage as we increase $k$ from 1 to $D$, where $D$ is a lower bound on the diameter of the largest strongly connected component of the dataset, computed using the NetworkX Hagberg et al. (2008). For each dataset, we observe a value of $k$ after which the efficiency does not improve, while still achieving the desired coverage. This behavior can suggest a heuristic akin to the 'elbow method' used in clustering analysis to determine the number of clusters for choosing the value of $k$ in NAPS.

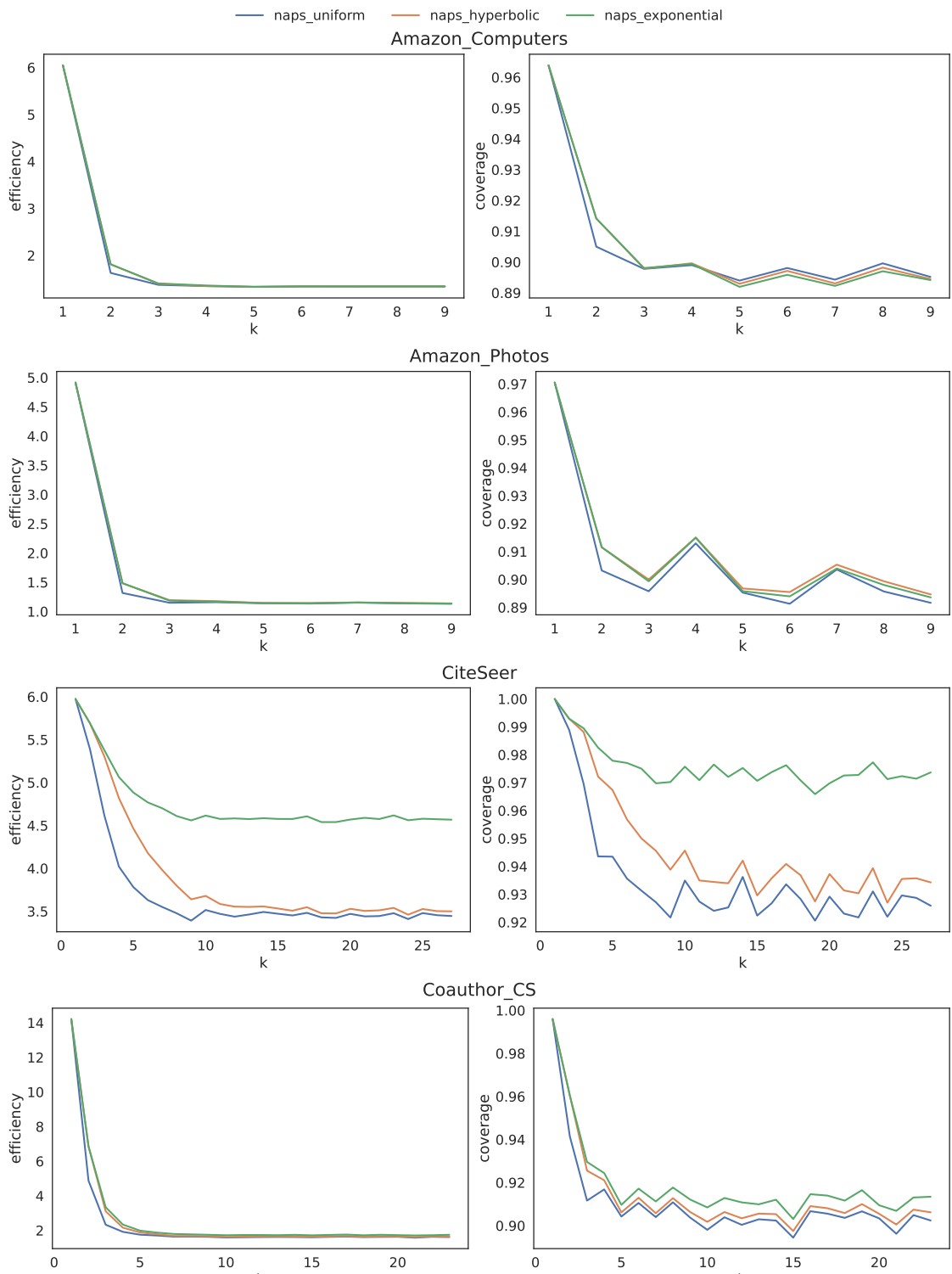

Figure C2: Plotting the Efficiency and Coverage when using NAPS for $k$ from 1 to $D$. The above results are with FS split and $\alpha = 0.1$.

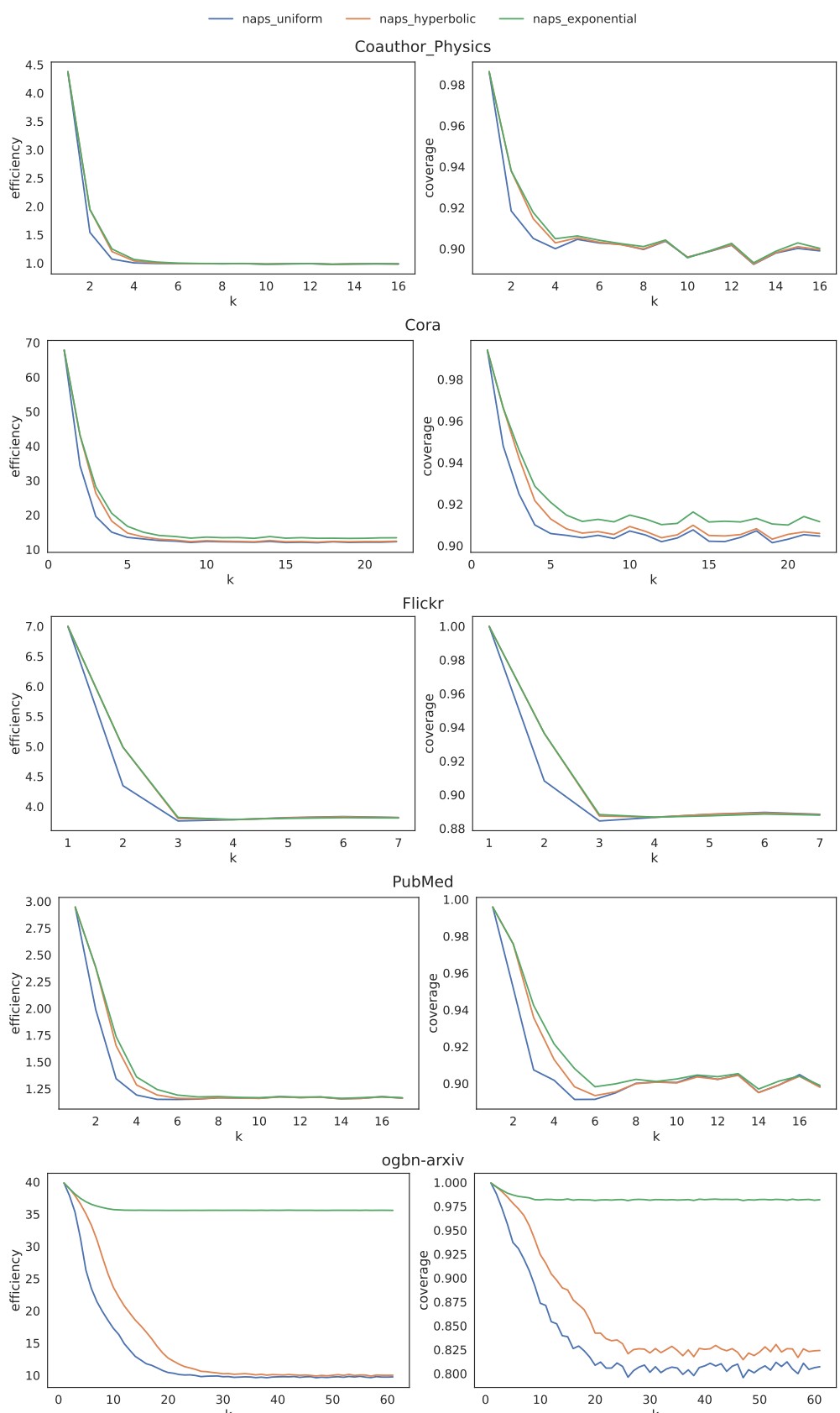

Figure C2: Plotting the Efficiency and Coverage when using NAPS for $k$ from 1 to $D$. The above results are with FS split and $\alpha = 0.1$ (cont.). The ogbn-products dataset is omitted due to size and lack of data points.

## C.3    Conformalized GNN

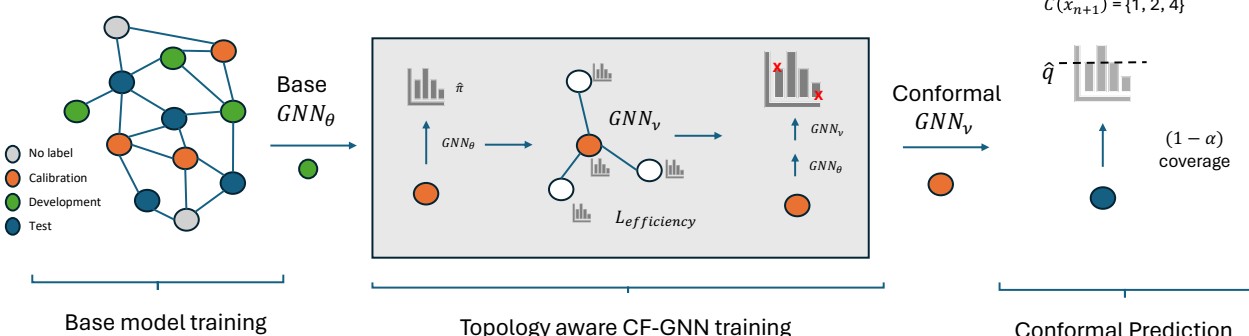

Figure C3: Procedure for training CFGNN. First (left), the base model is trained on the training set. Then, (middle) CFGNN is trained to maximize efficiency over the calibration set. Finally, (right) the non-conformity scores from the combined models are used to generate the prediction sets.

Huang et al. (2023) introduces CFGNN as a conformal prediction method specific for graphs. Figure C3 shows the end-to-end CFGNN procedure split into three steps. The first is to train a base GNN model, $GNN_\theta$, on the development nodes, $\mathcal{V}_{\text{dev}}$, normally using a label-based loss function such as Cross Entropy Loss. Using the outputs of $GNN_\theta$ as inputs, a second GNN, $GNN_\varphi$ is trained using an efficiency-based loss function proposed by Huang et al. (2023). Equation 19 presents the efficiency-based loss function for node classification, where $\sigma$ is the sigmoid function and $\tau$ is a temperature hyperparameter. The calibration nodes, $\mathcal{V}_{\text{calib}}$, are split into two sets, $\mathcal{V}_{\text{cor-cal}}$ and $\mathcal{V}_{\text{cor-test}}$ for training and validation, respectively. The fully trained $GNN_\varphi$ is then used for the conformal prediction and prediction set construction.

$$\hat{\eta} = \text{DiffQuantile}(\{s(\boldsymbol{x}_i, y_i)\}, (1-\alpha)(1 + 1/|\mathcal{V}_{\text{cor-cal}}|))$$

$$\mathcal{L}_{eff} = \frac{1}{|\mathcal{V}_{\text{cor-cal}}|} \sum_{i \in \mathcal{V}_{\text{cor-cal}}} \sum_{k \in \mathcal{Y}} \sigma\left(\frac{s(\boldsymbol{x}_i, k) - \hat{\eta}}{\tau}\right) \tag{19}$$

**Impact of Inefficiency Loss:** Figure C4 compares the efficiency of 'cfgnn_aps', 'cfgnn_orig,' and 'aps_randomized' for all the other datasets. For the FS split, we see that 'cfgnn_aps' improves upon 'cfgnn_orig' for all datasets and can improve upon 'aps_randomized' for most datasets. We see that CFGNN is still quite brittle for all datasets for the LC Split. CFGNN can still improve upon 'aps_randomized' for datasets with a larger number of classes, as seen with Cora and ogbn-products.

**Scaling CFGNN:** Expanding on Table 2 from Section 3, Table C1 present the runtime improvements (see Table C1a) as well as the efficiencies (see Table C1b) for each CFGNN implementation using the best CFGNN architecture found through hyperparameter tuning (see Section D4). We observe that our improved implementation achieves comparable efficiency to the original in only 50 epochs as opposed to 1000 used in the original.

Table C1: Impact of different CFGNN implementations between the original, batching, and batching+caching. Used the **best** CFGNN architecture (w.r.t. validation efficiency) for each dataset. We run 5 trials for each setup and report a 95% confidence interval. OOM = Out of Memory

(a) Impact on Runtime

| dataset($\downarrow$) / method($\rightarrow$) | original | batching | batching+caching |
|---|---|---|---|
| CiteSeer | $379.28 \pm 9.36$ | $36.36 \pm 0.97$ | $27.45 \pm 0.80$ |
| Amazon_Photos | $496.36 \pm 4.92$ | $102.92 \pm 0.95$ | $54.44 \pm 1.36$ |
| Amazon_Computers | $664.98 \pm 10.90$ | $205.29 \pm 3.96$ | $73.85 \pm 1.56$ |
| Cora | $1378.01 \pm 13.35$ | $203.61 \pm 2.52$ | $73.60 \pm 1.53$ |
| PubMed | $571.60 \pm 12.09$ | $219.63 \pm 5.23$ | $109.68 \pm 1.64$ |
| Coauthor_CS | $638.56 \pm 6.14$ | $88.49 \pm 1.14$ | $31.67 \pm 0.53$ |
| Coauthor_Physics | $5942.30 \pm 176.90$ | $3918.38 \pm 17.16$ | $1585.26 \pm 6.84$ |
| Flickr | $868.87 \pm 9.98$ | $567.39 \pm 6.50$ | $56.71 \pm 1.30$ |
| ogbn-arxiv | $410.91 \pm 8.29$ | $373.19 \pm 3.64$ | $111.38 \pm 1.95$ |
| ogbn-products | OOM | OOM | $8709.16 \pm 55.00$ |

(b) Impact on Efficiency

| dataset($\downarrow$) / method($\rightarrow$) | original | batching | batching+caching |
|---|---|---|---|
| CiteSeer | $2.52 \pm 0.27$ | $2.45 \pm 0.19$ | $2.42 \pm 0.37$ |
| Amazon_Photos | $1.06 \pm 0.01$ | $1.06 \pm 0.02$ | $1.12 \pm 0.02$ |
| Amazon_Computers | $1.29 \pm 0.05$ | $1.15 \pm 0.01$ | $1.14 \pm 0.01$ |
| Cora | $6.81 \pm 1.07$ | $8.34 \pm 1.12$ | $7.96 \pm 0.59$ |
| PubMed | $1.17 \pm 0.00$ | $1.16 \pm 0.00$ | $1.17 \pm 0.00$ |
| Coauthor_CS | $1.10 \pm 0.01$ | $1.15 \pm 0.01$ | $1.14 \pm 0.01$ |
| Coauthor_Physics | $0.97 \pm 0.01$ | $1.01 \pm 0.03$ | $0.99 \pm 0.01$ |
| Flickr | $4.23 \pm 0.07$ | $4.23 \pm 0.04$ | $4.24 \pm 0.03$ |
| ogbn-arxiv | $7.07 \pm 0.05$ | $7.28 \pm 0.06$ | $6.91 \pm 0.01$ |
| ogbn-products | OOM | OOM | $1.86 \pm 0.06$ |

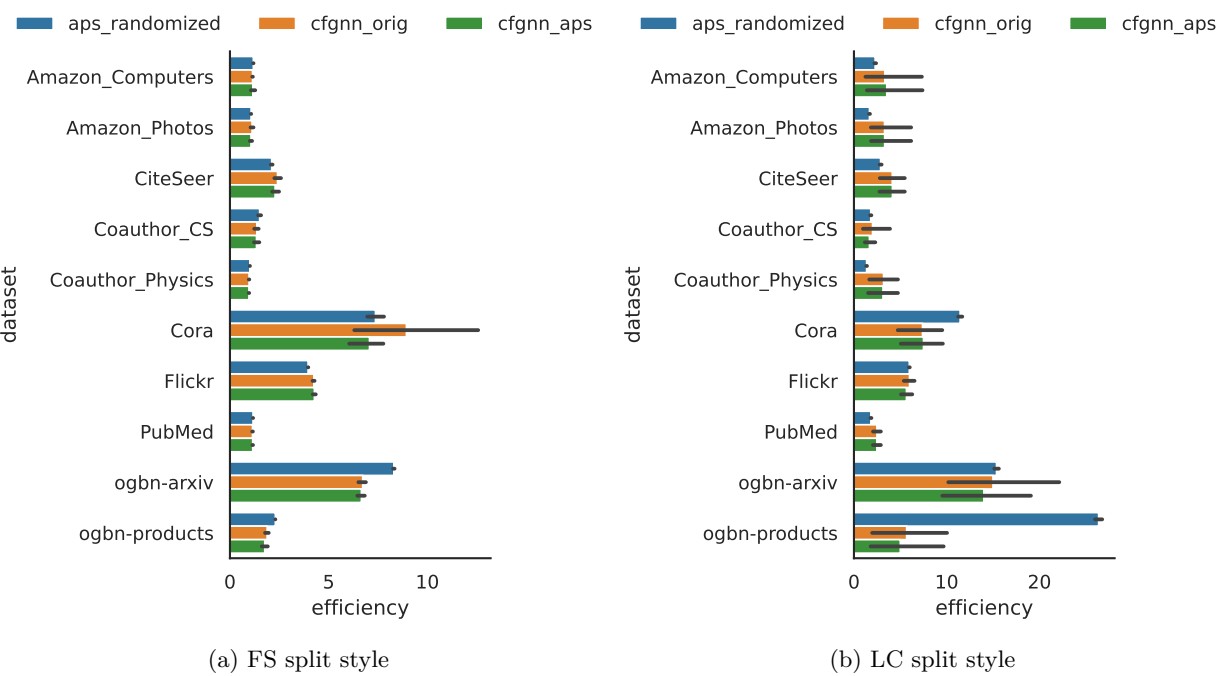

(a) FS split style

(b) LC split style

Figure C4: Bar charts denoting efficiency for 'cfgnn_aps', 'cfgnn_orig', and 'aps_randomized' for both split styles at $\alpha = 0.1$. We see that 'cfgnn_aps' improves or matches efficiency in most cases.

# D    Datasets and Hyperparameter Tuning Details

## D.1    Datasets

We selected datasets of varying sizes and origins to evaluate the performance of the graph conformal prediction methods. The first category of datasets is citation datasets, where the nodes are publications and the edges denote citation relationships. Nodes have features that are bag-of-words representations of the publication. The task is to predict the category of each publication. The citation networks we use are **CiteSeer** (Yang et al., 2016), CoraFull (Shchur et al., 2018), an extended version of the common Cora (Yang et al., 2016) citation network dataset, and **Pubmed** (Yang et al., 2016). The second category comes from the Amazon Co-Purchase graph  (McAuley et al., 2015), where nodes represent goods, edges represent goods frequently bought together, and node features are bag-of-words representations of product reviews. The task is to predict the category of a good. We use the **Amazon_Photos** and **Amazon_Computers** datasets. The last category is co-authorship networks extracted from the Microsoft Academic Graph (Wang et al., 2020) used for KDD Cup'16. The nodes are authors, edges represent coauthorship, and node features represent paper keywords of the author's publications. The task is to predict the author's most active field of study. We use **Coauthor_CS** and **Coauthor_Physics**, which both come from the Microsoft Academic Graph (Wang et al., 2020). Other datasets that were use include **Flickr** (Zeng et al., 2020), **ogbn-arxiv**, and **ogbn-products** Hu et al. (2020). These last three datasets have predefined splits for train/validation/test, shown in Table D2, which we used when constructing our train/validation/calibration/test splits for the different split styles. We used the version given by the Deep Graph Library  (Wang et al., 2019) for all the datasets. DGL uses an Apache 2.0 license, and OGB uses an MIT license.

Table D1 presents summary statistics for each dataset. These include the average local clustering coefficient (Avg CC), global clustering coefficient (Global CC) (Newman, 2018), an approximate lower bound on the diameter ($D$) given by Magnien et al. (2009), node homophily ratio ($\hat{H}$) (Pei et al., 2020), and expected node homophily ratio ($H_{rand}$). Figure D1 shows the label distribution for each dataset.

Table D1: Summary statistics for all datasets evaluated.

| Dataset | Nodes | Edges | Classes | Features | Avg CC | Global CC | $D$ | $\hat{H}$ | $H_{rand}$ |
|---|---|---|---|---|---|---|---|---|---|
| CiteSeer | 3,327 | 9,228 | 6 | 3,703 | 0.141 | 0.130 | 28 | 0.722 | 0.178 |
| Amazon_Photos | 7,650 | 238,163 | 8 | 745 | 0.404 | 0.177 | 10 | 0.836 | 0.165 |
| Amazon_Computers | 13,752 | 491,722 | 10 | 767 | 0.344 | 0.108 | 10 | 0.785 | 0.208 |
| Cora | 19,793 | 126,842 | 70 | 8,710 | 0.261 | 0.131 | 23 | 0.586 | 0.022 |
| PubMed | 19,717 | 88,651 | 3 | 500 | 0.060 | 0.054 | 17 | 0.792 | 0.357 |
| Coauthor_CS | 18,333 | 163,788 | 15 | 6,805 | 0.343 | 0.183 | 24 | 0.832 | 0.112 |
| Coauthor_Physics | 34,493 | 495,924 | 5 | 8,415 | 0.378 | 0.187 | 17 | 0.915 | 0.321 |
| Flickr | 89,250 | 899,756 | 7 | 500 | 0.033 | 0.004 | 8 | 0.322 | 0.267 |
| ogbn-arxiv | 169,343 | 1,166,243 | 40 | 128 | 0.118 | 0.115 | 62 | 0.567 | 0.077 |
| ogbn-products | 2,449,029 | 61,859,140 | 47 | 100 | 0.411 | 0.130 | 27 | 0.817 | 0.106 |

Table D2: Predefined splits from original source noted.

| Dataset | # Train | # Valid | # Test |
|---|---|---|---|
| Flickr | 44,625 | 22,312 | 22,313 |
| ogbn-arxiv | 90,941 | 29,799 | 48,603 |
| ogbn-products | 196,615 | 39,323 | 2,213,091 |

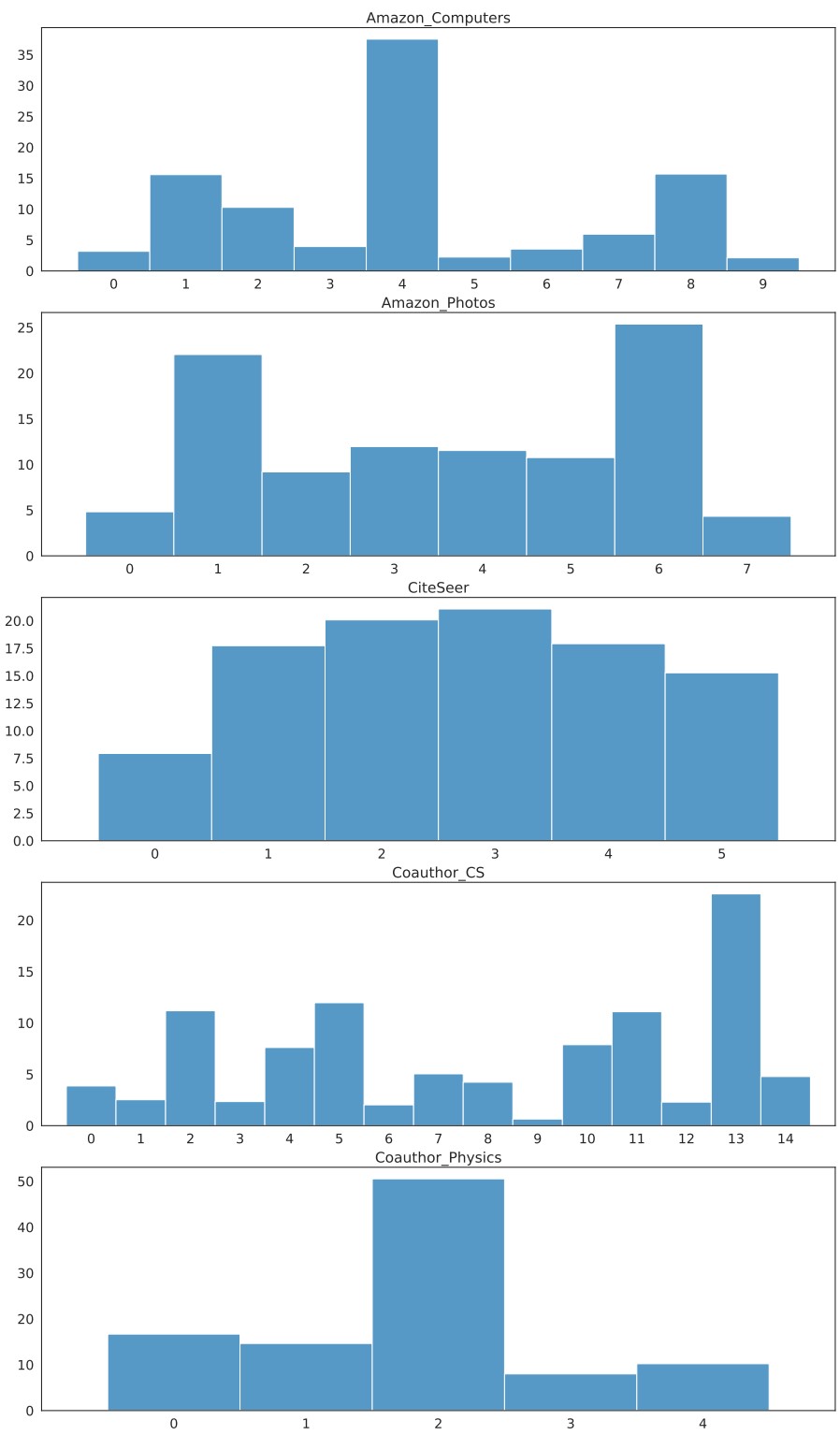

Figure D1: Plots of label distribution for each dataset. Each class for each dataset has at least one occurrence.

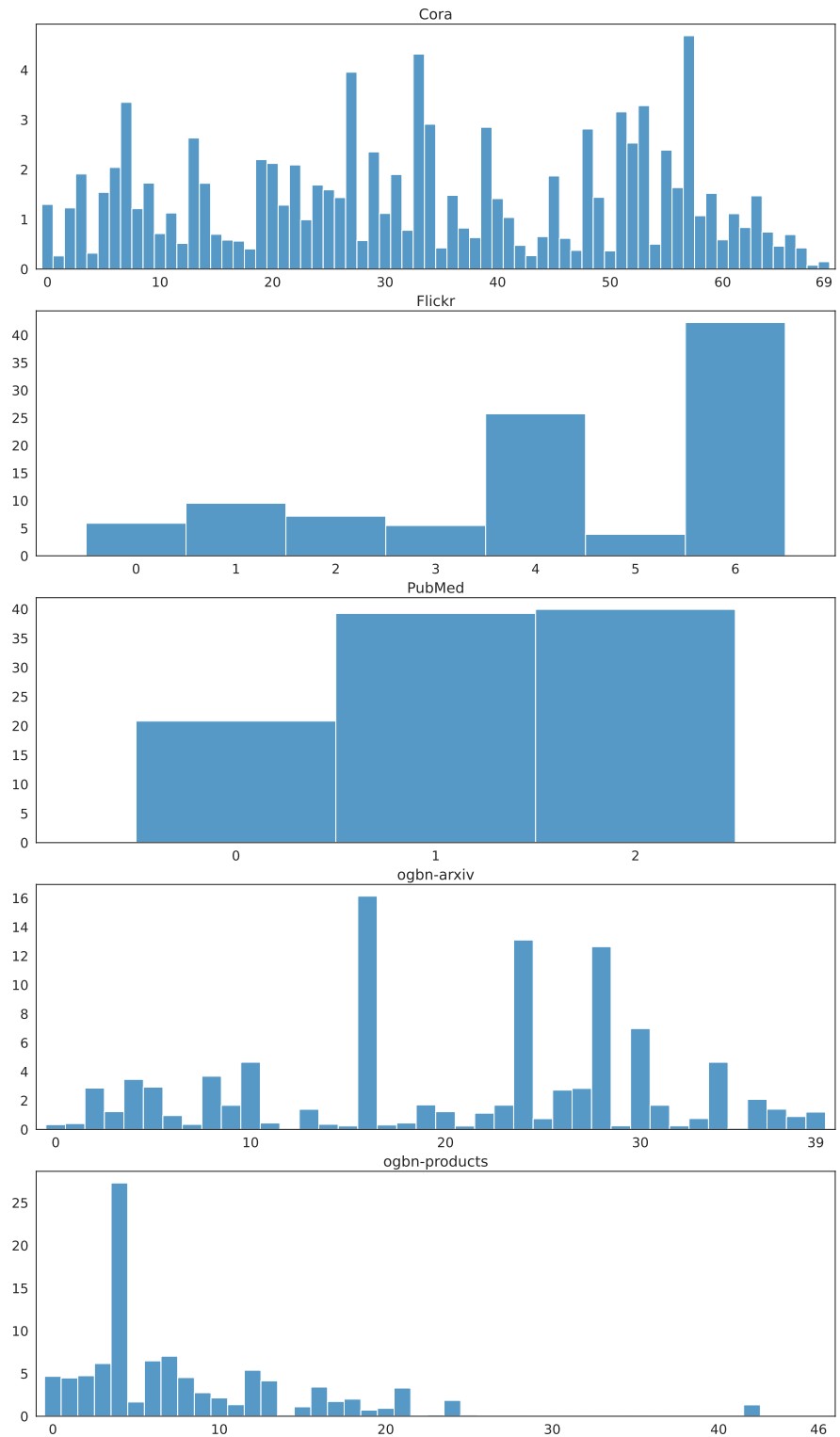

Figure D1: Plots of label distribution for each dataset. Each class for each dataset has at least one occurrence (cont.).

### D.2 Hyperparameter Tuning

Hyperparameter tuning was done using Ray Tune (Liaw et al., 2018). The hyperparameters for the base model were tuned via random search using Table D3 for each model type (e.g., GCN, GAT, and GraphSAGE), for each dataset, except for the OGB (Hu et al., 2020) datasets, and each splitting scheme (FS vs LC and different value settings). For the OGB datasets, we took the hyperparameters and architectures from the corresponding leaderboard for all splitting schemes.

Once we got the best base model for each dataset and splitting scheme, we tune the hyperparameters for the CF-GNN model via random search using Table D4 for each model type (e.g., GCN, GAT, and GraphSAGE), for each dataset[4]. For the FS splitting schemes, we enfore $|\mathcal{D}_{\text{calib}}| = |\mathcal{D}_{\text{test}}| = (1 - |\mathcal{D}_{\text{train}}| - |\mathcal{D}_{\text{valid}}|)/2$. For the ogbn-products dataset, due to its size, we used a batch size of 512 for the CF-GNN training and also used a NeighborSampler with fanouts $[10, 10, 5]$ rather than a MultiLayerFullNeighborSampler.

All experiments with the ogbn-products datasets were run on a single A100 GPU, while the remaining experiments for the other datasets were run on a single P100 GPU.

Table D3: Hyperparameter search space for the base GNN model for non-OGB datasets. The last two rows are layer-type specific for GAT and GraphSAGE, respectively.

| Hyperparameter | Search Space |
| --- | --- |
| batch_size | 64 |
| lr | $\text{loguniform}(10^{-4}, 10^{-1})$ |
| hidden_channels | $\{16, 32, 64, 128\}$ |
| layers | $\{1, 2, 4\}$ |
| dropout | $\text{uniform}(0.1, 0.8)$ |
| heads | $\{2, 4, 8\}$ |
| aggr_fn | $\{\text{mean, gcn, pool, lstm}\}$ |

Table D4: Hyperparameter search space for the CF-GNN model. The last two rows are layer-type specific for GAT and GraphSAGE, respectively.

| Hyperparameter | Search Space |
| --- | --- |
| batch_size | 64 |
| lr | $\text{loguniform}(10^{-4}, 10^{-1})$ |
| hidden_channels | $\{16, 32, 64, 128\}$ |
| layers | $\{1, 2, 3, 4\}$ |
| dropout | $\text{uniform}(0.1, 0.8)$ |
| $\tau$ | $\text{loguniform}(10^{-3}, 10^{1})$ |
| heads | $\{2, 4, 8\}$ |
| aggr_fn | $\{\text{mean, gcn, pool, lstm}\}$ |

---

[4]The configuration files for all the experiments and best architectures are available in our codebase: `https://github.com/pranavmaneriker/graphconformal-code`.

# E   Additional Empirical Results, Analysis, and Insights

This section expands the figures and tables in the main body for all datasets considered. Figure E1 compares TPS and TPS-Classwise for all the datasets. We observe that TPS-Classwise achieves the desired label-stratified coverage of 0.9 while TPS doesn't necessarily for the FS split. For the LC split, we observe that TPS-Classwise slightly improves on TPS in terms of label-stratified coverage; however, neither method necessarily achieves the target label-stratified coverage. In Figure E2, we find TPS-Classwise generally is less efficient than TPS for FS and LC splitting. Figure E3 compares the label-stratified coverage for APS with and without randomization. We observe that randomization does sacrifice the label-stratified coverage for both FS and LC splitting. Noticeably, the change in coverage is smaller for the LC split. Figure E4 compares the efficiency of APS with and without randomization at $\alpha = 0.1$. For both split types (FS & LC), we observe that the randomized version of APS produces more efficient prediction sets, in line with Theorem 2.3. The efficiency improvements come with a sacrifice in label-stratified coverage since smaller prediction set sizes are preferred over covering every class, particularly if the classes are rare. To visualize this trade-off, we observe that in both Figure E1 and E3 the difference in label stratified coverage for ogbn-products with FS splitting is more extreme than with other datasets and with LC splitting. This is because ogbn-products has a lot of classes that have almost no representation (see Figure D1). Datasets with near-uniform label distribution (e.g., PubMed, CiteSeer) – or when using LC split, which controls for label counts – we observe that label-stratified coverage isn't sacrificed as much in the name of efficiency.

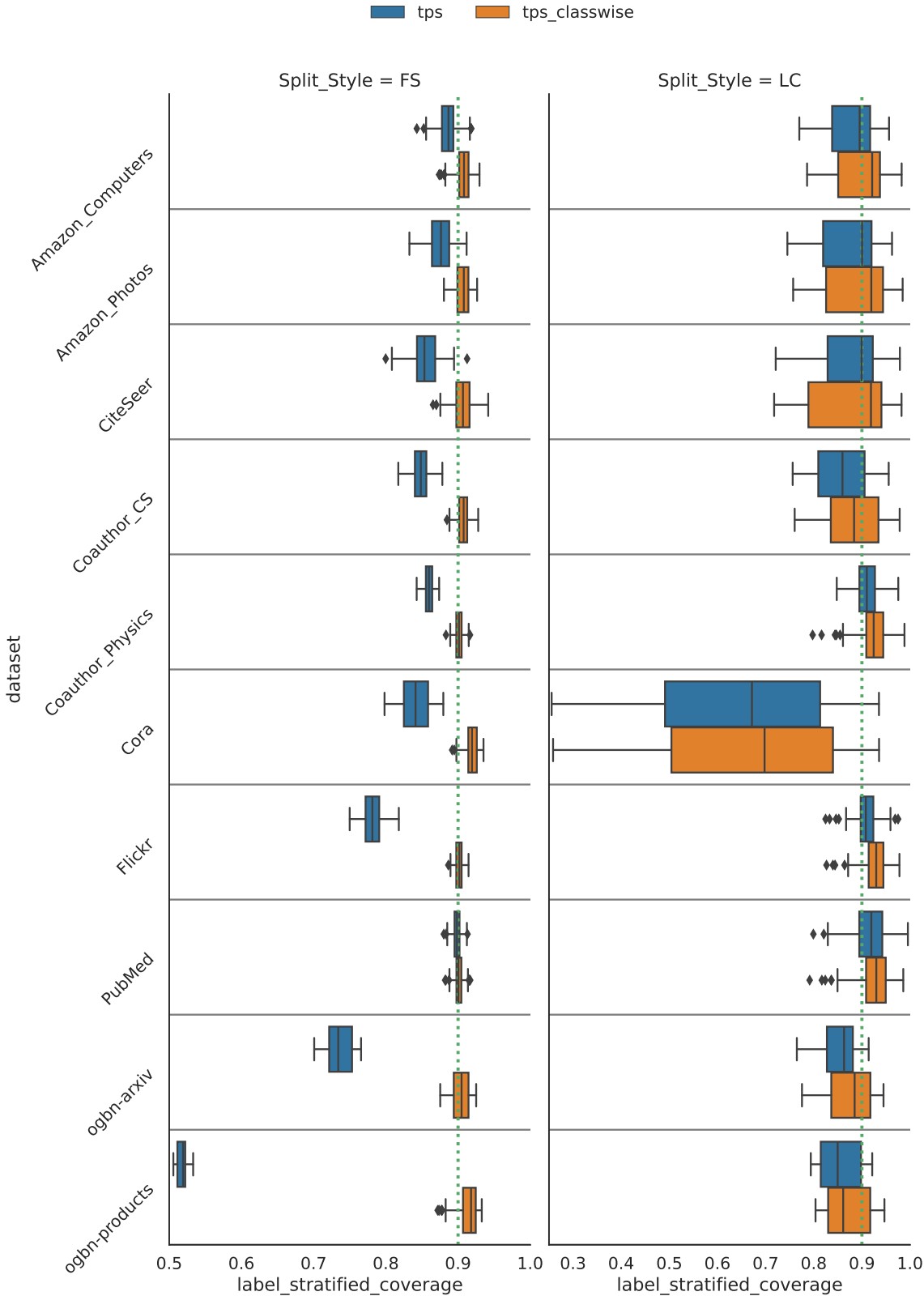

Figure E1: Plots for TPS vs TPS-Classwise for all the data sets at $1 - \alpha = 0.9$ coverage (green dotted line). TPS-Classwise, on average, meets label-stratified coverage for FS split (left). For the LC split, the label stratified coverage slightly improves with TPS-Classwise but does not necessarily meet the $1 - \alpha$ coverage.

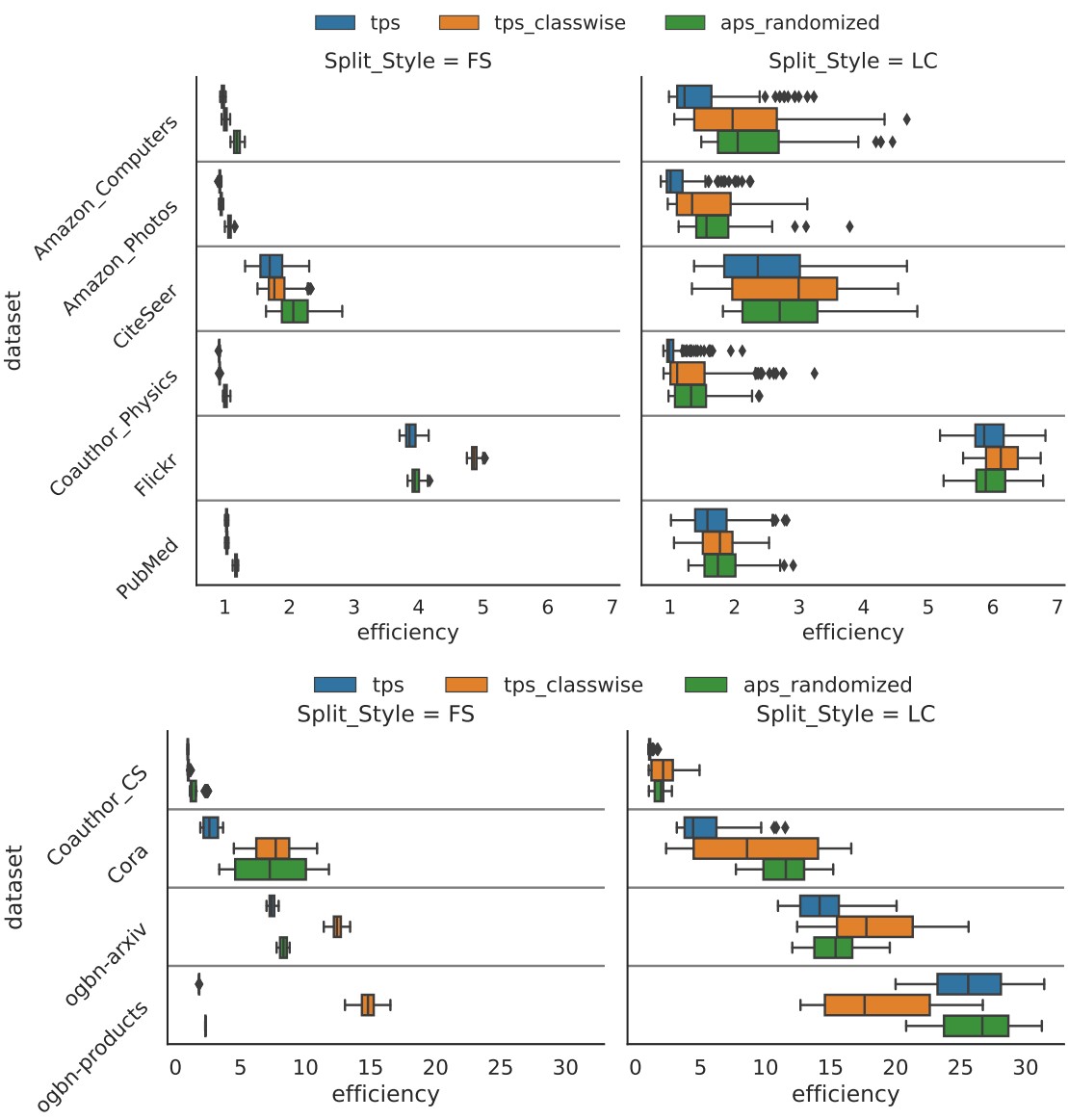

Figure E2: Plot for TPS vs TPS-classwise at $\alpha = 0.1$. For the FS split type, TPS-classwise becomes more inefficient compared to TPS for larger graph sizes, but is competitive in other settings. To maintain label-stratified coverage, TPS-classwise may be forced to overcover certain classes at the cost of efficiency.

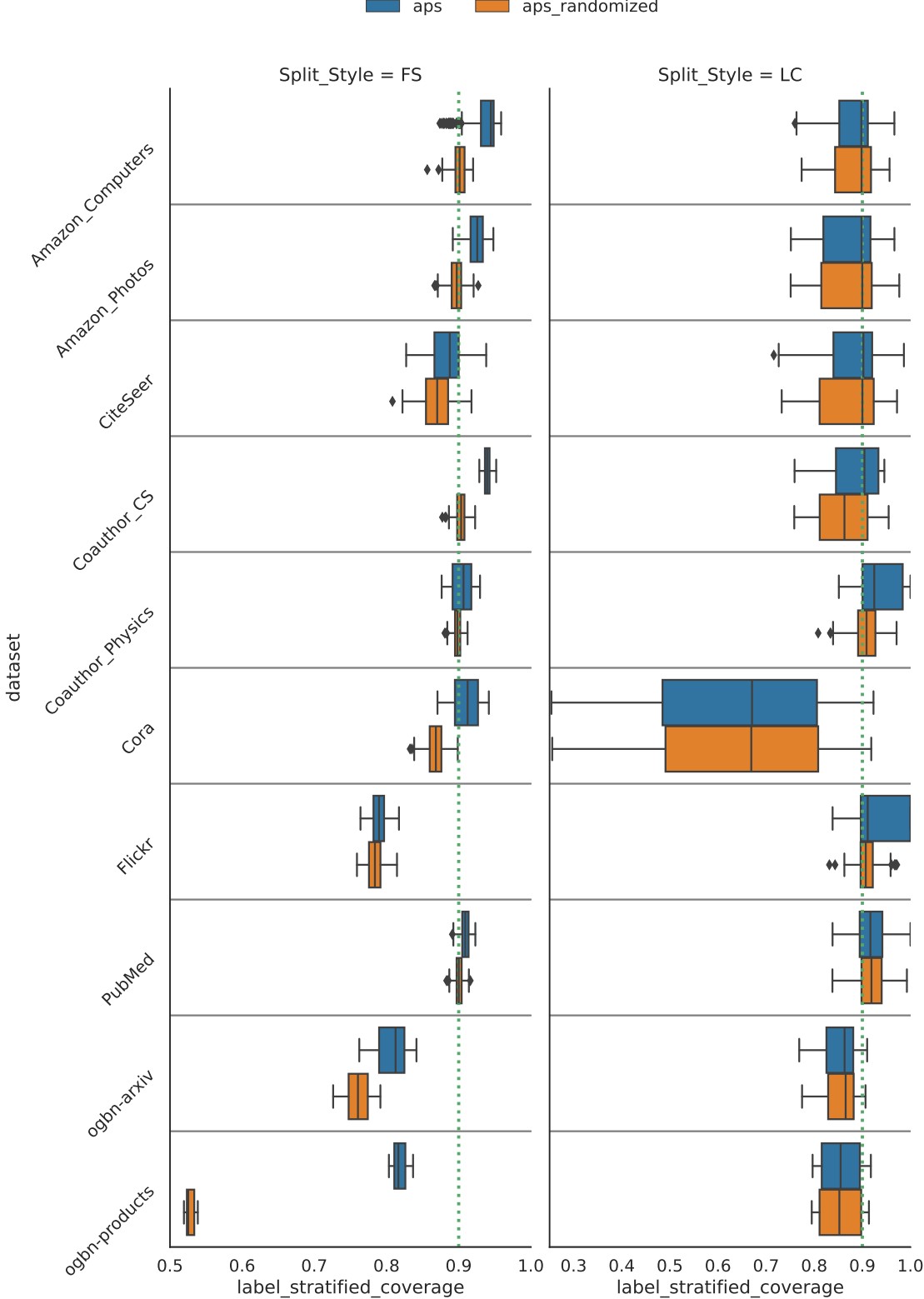

Figure E3: Plots for APS vs APS Randomized for all the data sets at $1 - \alpha = 0.9$ coverage (green dotted line). For the FS split type, APS-Randomized has a lower label-stratified coverage. However, with the LC split type, the decrease in label-stratified coverage is not as significant.

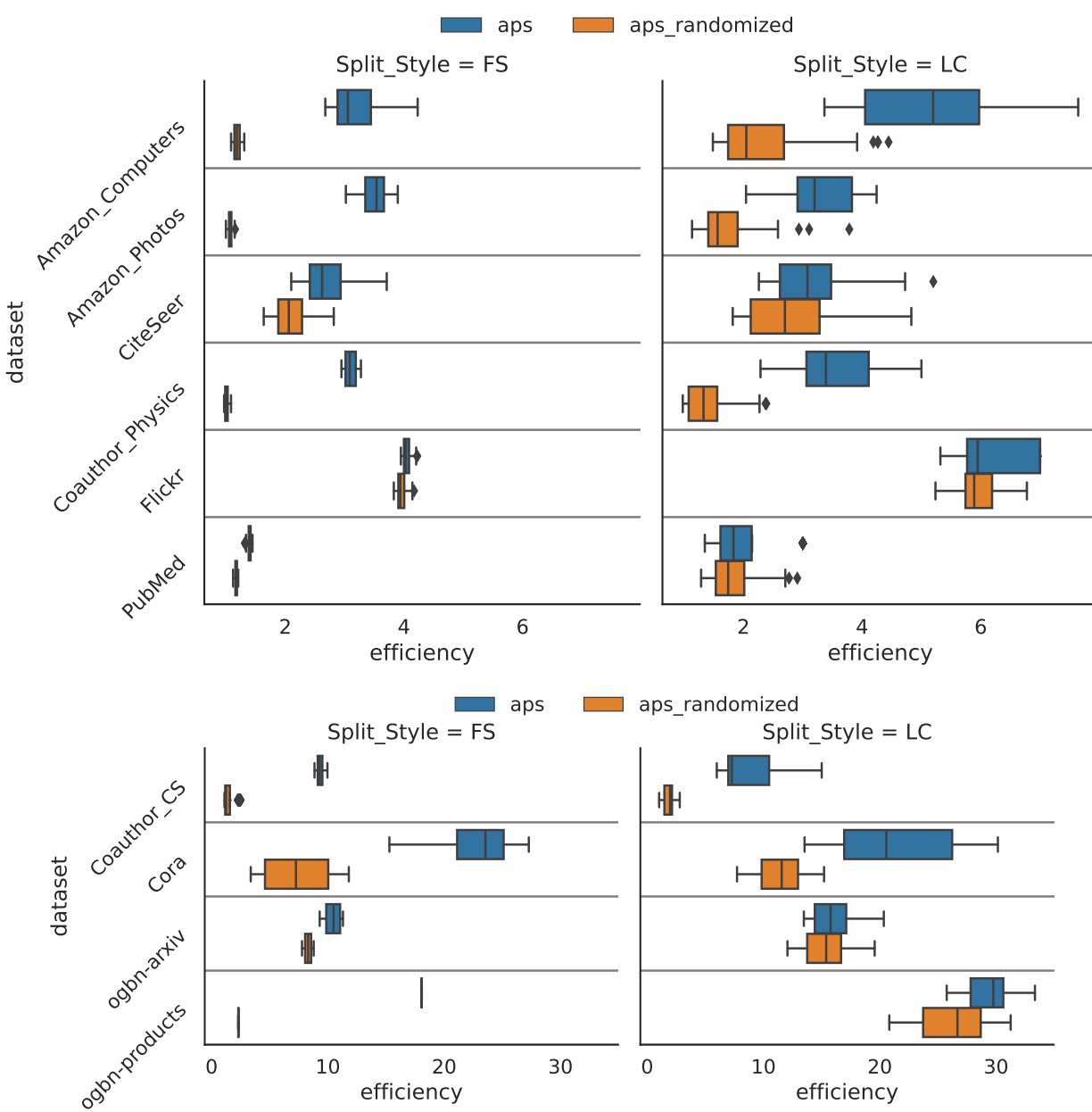

Figure E4: Plots for APS vs APS-randomized at $\alpha = 0.1$. For both split types (FS & LC), the randomized version of APS produces more efficient sets for all the datasets.

### E.1 Overall Results

In Figure E5, we provide a plot of all the different methods discussed in this work for each dataset across different values of $\alpha$. If applicable, for each method, we show the best-performing version, e.g., APS with randomization vs without and CFGNN with APS training ('cfgnn_aps') rather than TPS training ('cfgnn_orig'). We present the results for $k = 5$ for the different NAPS variations, since almost all datasets – except for CiteSeer and ogbn-arxiv – achieved their best efficiencies at or before that point.

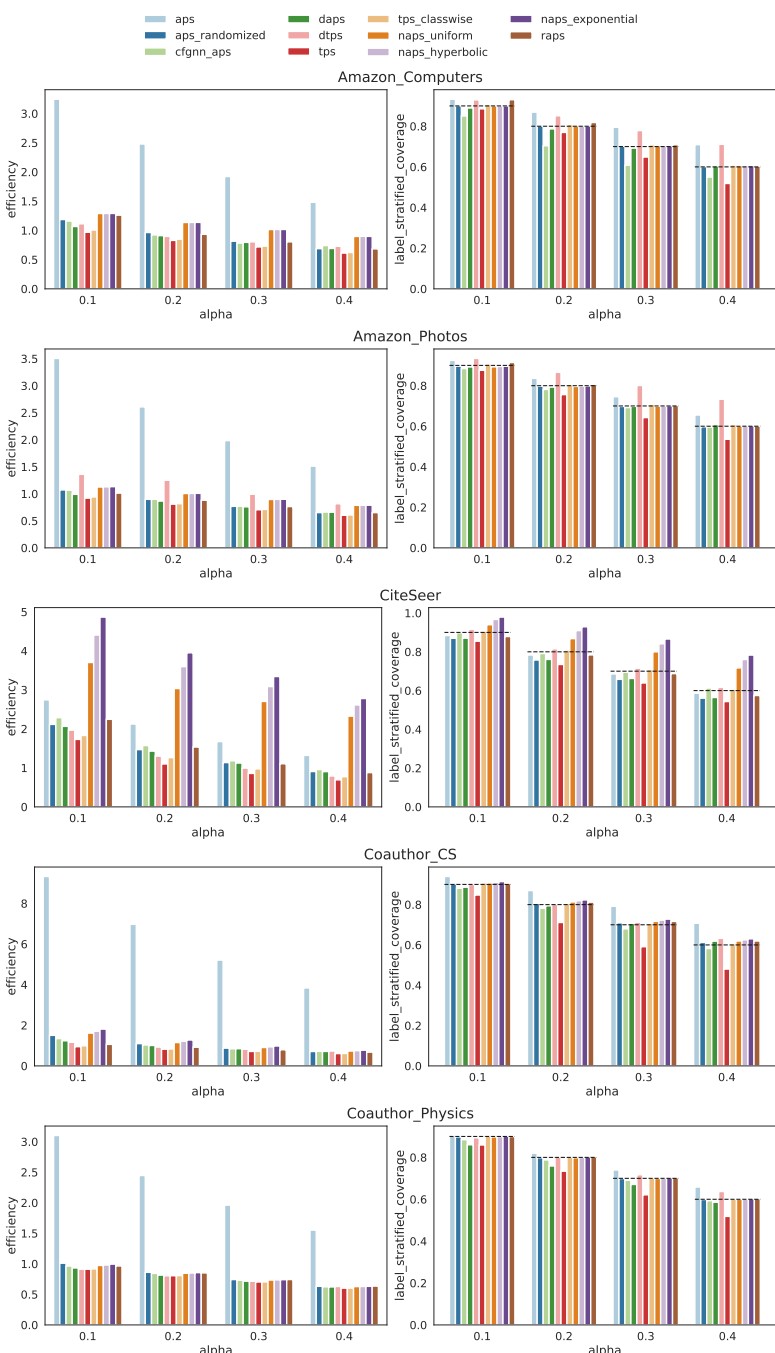

Figure E5: Plots for efficiency vs $\alpha$ for all the major methods (with best parameters) across all the datasets (cont.). Among the baseline methods, TPS consistently has the best efficiency. The results are for the FS split style. The dashed black line indicates the desired label-stratified coverage.

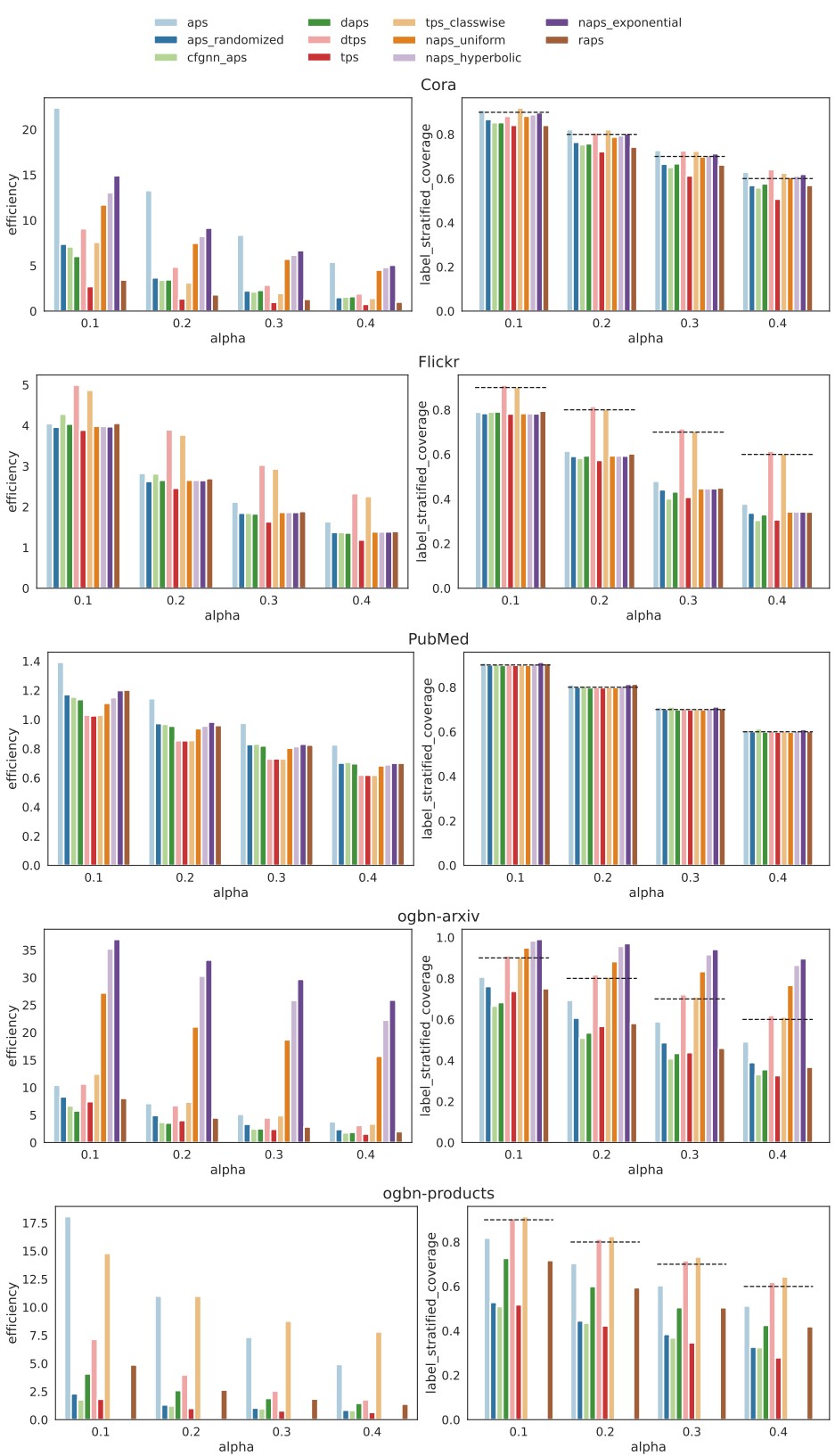

Figure E5: Plots for efficiency vs $\alpha$ for all the major methods (with best parameters) across all the datasets (cont.). Among the baseline methods, TPS consistently has the best efficiency. Results are for the FS split style. The dashed black line indicates the desired label-stratified coverage. NAPS results for the ogbn-products dataset are omitted due to size and lack of data points with existing hardware.

## F    Related Works

In this work, we focus on the prominent methods of graph conformal prediction for the node classification task under a transductive setting. These included standard conformal prediction methods, like TPS and APS, and graph-specific methods, like DAPS and CFGNN. Some works consider other graph-based scenarios and tasks. Within node classification, we can also consider the inductive setting, where nodes/edges arrive in a sequence, thus violating the exchangeability assumption. One such method is Neighborhood Adaptive Prediction Sets (NAPS), which we consider a transductive variation in this work. Beyond node classification on static graphs, very recently, there has been some work in link prediction (Zhao et al., 2024; Marandon, 2024). Our work can potentially be adapted to these recent ideas (especially Zhao et al. (2024)), although they make additional assumptions beyond the scope of the current work. Additionally, how these ideas would expand for dynamic graphs would be an interesting future direction to pursue (Davis et al., 2025).

Beyond conformal prediction, many works propose ways to construct model-agnostic uncertainty estimates for classification tasks (Abdar et al., 2021; Guo et al., 2017; Kull et al., 2019). There are also methods specific to GNNs (Hsu et al., 2022; Wang et al., 2021) that leverage network principles (e.g., homophily). However, unlike conformal prediction, these methods can fail to provide the desired coverage guarantees. For an empirical comparison of popular UQ methods for GNN, please reference Huang et al. (2023).

