# OpenReview forum: "Conformal Prediction: A Theoretical Note and Benchmarking Transductive Node Classification in Graphs"
_TMLR — Accepted by TMLR_

### Review · Reviewer_JtiA · 2025-02-10

**Summary Of Contributions:**

The paper investigates conformal prediction for the node classification task in graphs, focusing on the Split-CP technique. The paper makes the following contributions:

(1)	The paper provides a theoretical result that clarifies sufficient conditions to guarantee that one score function will produce a more efficient prediction set than another score function.

(2)	The paper discusses Threshold Prediction Sets (TPS), focusing on the issue of adaptability. The paper contrasts the standard version of TPS with a classwise version (TPS-classwise) that provides coverage for each label at the potential cost of increased variance. The paper experimentally compares the performance of TPS and TPS-classwise under Full-Split and Label-Count sample partitioning, showing that for some datasets TPS-classwise can provide improved adaptability in the Full-Split sample partitioning, especially in cases where the number of samples from each class can vary substantially.

(3)	The paper discusses Adaptive Prediction Sets (APS), focusing on the issue of randomized versus deterministic non-conformity scores. The paper provides experimental results showing that randomized APS consistently achieves better efficiency than deterministic APS.

(4)	The paper investigates a conformalized GNN (CFGNN), (i) replicating an experiment from a past work that used deterministic APS and showing that the efficiency improvement effectively disappears if randomized APS is used instead; (ii) showing that a small but consistent performance improvement can be achieved if randomized APS is used during calibration; (iii) demonstrating that the CFGNN’s performance in the Label-Count sample partitioning setting is considerably worse; and (iv) reporting on a scalable implementation using batching and caching.

**Audience:**

Yes

**Claims And Evidence:**

No

**Requested Changes:**

(1)	It’s important that the claims of the paper match what is demonstrated by experiment and theory. Currently the claims are relatively vague and general, and they are not completely supported. Please rewrite the final paragraph of the introduction (and revisit the abstract) to ensure that there is a match between the summary description (claims) and the provided evidence. Most importantly, the paper only presents results for node classification. This should be very clearly specified in the abstract and introduction. Currently, the paper just talks about “graph conformal prediction”, but it does not provide any evidence to support its claims for graph level tasks. Aside from this, the paper claims to provide a “deeper theoretical understanding of some design choices”. In terms of theory, the paper really provides only one theoretical result that clarifies sufficient conditions for when one score function will outperform another.

Finally, the paper claims to provide “intuition on when and how to evaluate CP for graph data”. It is difficult to match this claim to a specific part of the text. I presume that Section 3.4 is supposed to serve this purpose. But it does not really provide clear recommendations concerning “when and how”. Currently it is a collection of relatively random observations that do not form a coherent whole. After reading this, and being presented with a graph dataset, I would really struggle to know what the authors suggest I do and why. Please rewrite Section 3.4 to make the conclusions and practical guidelines much clearer.

(2)	Theorem 3.1 is currently very difficult to understand – both in terms of the proof and its application.

(a)	How are we supposed to interpret “significance level”? My understanding of this is that it is the level at which one accepts that something is statistically significant – hence reporting such as “at the 5 percent significance level”. This is decided by the researcher/scientist as a threshold. However, Theorem 3.1 involves score-specific “significance levels” – am I to interpret $\alpha_c$ as a function of the score function and the data/distribution? Is it really then a significance level?

(b)	Is it clear that “the significance level $\alpha_c^A$” should be common to all incorrect classes? This confusion is related to my struggle to understand how the significance level is defined. Later, there appears to be a sum over classes, which is simplified to be a multiplication by (K-1). But if this is a probability defined for a specific incorrect class (and if it is not, then I don’t understand why there is a sum), then it is difficult to see why it should be the same for every class. Class 1 may be very easy to distinguish, so it is never incorrectly included in the set. On the other hand, class 2 may be very similar to 3 other classes, so it is often incorrectly included. The probabilities (and hence “significance levels”) would therefore have to be different for these classes, because you are providing both upper and lower bounds.

(c)	In the proof of Theorem 3.1, for the step to K classes, there is an appearance of $y_i$. What is $i$ in this equation? For the two class case, it is $y_{n+1}$. Why does it change to $i$? If it is an application of exchangeability, then that is fine, but there is a need to at least specify what $i$ is.

(d)	In the proof of Theorem 3.1, there is a sum over $y^’_i$ - what is the nature of this sum? It doesn’t seem to make sense that it is over $i$. But there is no other indexing variable. Based on the subsequent appearance of $K-1$, it would seem that it is a sum over the incorrect classes. But the notation does not correctly indicate that it is a sum over classes. Please fix the notation.

(e)	The intended application of this theorem is hard to understand. Again, it may all stem from my difficulty in understanding the definition of a “significance level”. But, as best as I can interpret it, the significance level for each score function is being determined from the data. These are then compared, and one can judge whether the prediction set is more efficient. But at this point, the prediction sets are themselves available, so one can directly judge whether one is more efficient than the other. What is the value of the theorem?

(3)	The caption of Figure 3 and the associated description in the text are hard to understand. Please rewrite both (the caption and the paragraph) so that it is clear what is being shown and what point is being made by the figure. In its current form, it is hard to see the support for the claim that “This demonstrates the scenario that is a condition for Theorem 3.1”.

Minor Points:

M1. What does the following sentence mean (Section 3.4)?: “This particular dataset comes at the cost of efficiency, though this is not the case with all datasets.” How can a dataset “come at the cost of efficiency”? Please rephrase so that it is easier to understand.

M2. I don’t understand the following sentence in the caption of Figure 1: “For Labeled Stratified Coverage, TPS-Classwise can provide comparable performance to TPS.” All the figures are labelled as “label-stratified coverage”. It is very clear that the performance is not comparable for ogbn-arxiv under the Full-Sample split.

M3: Please number more equations – preferably all. None of the equations in the proof of Theorem 3.1 are numbered. This makes it challenging for me as a reviewer to ask questions or point to issues. It also makes it difficult for readers who may wish to build upon your work and refer, either amongst themselves or in writing, to specific steps or developments.

M4. Typo: “adapatability” – p4.

**Strengths And Weaknesses:**

Strengths:

S1. The paper is well-written and provides a clear summary of conformal prediction for graphs, focusing on the task of node classification.

S2. The paper presents several experimental results that provide insight into the behaviour of conformal prediction in graph learning.

S3. The paper describes a useful scalable implementation (and makes code available).

Weaknesses

W1. There is a concern that the claims (description of what the paper provides, at the end of the introduction and in the abstract) are too vague and are thus not clearly supported by the evidence, either experimental or theoretical, later in the paper.

W2. Although the theoretical result is a welcome contribution, it is currently presented in a confusing fashion. Due to confusion about key definitions, it is not clear to me what the theorem statement means and whether the proof is correct. Aside from this, I struggle to understand the utility of the theorem, as detailed below in “Requested Changes”.

---

> ### Author Response · Authors · 2025-02-24
> **Response to Reviewer**
>
> We thank the reviewer for taking the time to review our paper.
>
> > [W1] "There is a concern that the claims (description of what the paper provides) are too vague and are thus not clearly supported by the evidence..."
>
> > > [RC1a] "Please rewrite the final paragraph of the introduction (and revisit the abstract) to ensure that there is a match between the summary description (claims) and the provided evidence."
>
> We will update the final paragraph of the introduction to be a separate subsection titled **Key Contributions**. In this section, we will note the following 3 contributions:
>
> 1. We perform a comprehensive analysis of the existing graph split-CP literature (e.g. CFGNN, DAPS) for node classification. This analysis aims to understand the choices made in method implementation (i.e., scalability) and method evaluation (i.e., dataset preparation, and baseline selection). We further expand our analysis to include the broadly applicable CP methods the literature discusses and uses like TPS and APS.
>
> 2. We provide a theorem for evaluating different conformal prediction methods in terms of efficiency (i.e., set size), _prior_ to deployment.
>
> 3. We created a Python library that captures the different parts of our analysis that practitioners can use when implementing and evaluating conformal prediction for graph data. We further implemented the methods to be scalable for large graphs, particularly for computationally expensive methods like CFGNN.
>
> > > [RC1b] "..the paper only presents results for node classification...the paper just talks about “graph conformal prediction”..."
>
> We state in the paper that our work focuses on (transductive) node classification but can make it very explicit in the abstract. Moreover, for graph-level tasks such as graph classification, the graph as a whole is treated as a single data point, similar to classification with images and tabular data. Thus, standard CP methods, which have been well-studied even beyond this work, can be applied
>
> > > [RC1c] "In terms of theory, the paper really provides only one theoretical result that clarifies sufficient conditions for when one score function will outperform another."
>
> Theorem 3.1 formalizes the importance of randomization for APS (in a general setting), which is often stated, but we could not find clear proof in the existing literature. Indeed, it is essential to make this point, especially in light of recent efforts (e.g., CFGNN that did not examine APS with randomization as a baseline). The theorem’s practical utility is that $\alpha_c^A$ can be computed using the calibration set, so which method is more efficient (i.e. smaller set size) can be determined, apriori.
>
> > > [RC1d] "Please rewrite Section 3.4 to make the conclusions and practical guidelines much clearer."
>
> We will update the paper to include clear practical guidelines with considerations such as whether computational resources are available or what is being optimized for (i.e. efficiency or adaptability). Although these findings are stated in the paper, we will summarize all of them for ease of utility.
>
> > [W2] The theoretical result is presented in a confusing fashion and the utility of the theorem is not clear
>
> > > [RC2] "Theorem 3.1 is currently very difficult to understand – both in terms of the proof and its application."
>
> (a) To be consistent with the conformal prediction literature and avoid using the term “significance level”, we will instead use “miscoverage level".
>
> (b) Following the exchangeability assumption of conformal prediction, we have that the pairs $\\{x_i, \tilde{y}\\}_{i = 1}^{n}$ for a **_fixed_** label, $\tilde{y}\in\mathcal{Y}$ are exchangeable. Theorem 3.1 relies on this fact, which we will explicitly make in the updated version. Because of this assumption, we can sample a random incorrect class for $x_i$ and compute $\alpha_c^A$ using those points.
>
> (c) We thank the reviewer for pointing this out and agree that it should be $y_{n + 1}$ and not $y_i$. We have fixed this for the updated version.
>
> (d) The reviewer is correct that the sum is taken over the incorrect labels. To make this point clear we’ll have the sum be over $y’ \in \mathcal{Y} \setminus \\{y_{n+1}\\}$ rather than just $y_i’$
>
> (e) To reiterate from RC1c, the theorem’s main utility is that a practitioner can determine which method is more efficient **before** deployment using the calibration set. This theorem applies to conformal prediction in general and is not tied to graphs.
>
> > > [RC3] Please rewrite both (the caption and the paragraph) so that it is clear what is being shown and what point is being made by the figure.
>
> We have updated the caption for Figure 3 to be more concise and clarify the relationship between the figures and Theorem 3.1. Particularly that the sufficient condition for Theorem 3.1 -- $\alpha_c^A - \alpha_c^{\tilde{A}}\geq\frac{2}{n + 1}$ -- is demonstrated by the vertical lines in the right plot of Figure 3.

---

### Review · Reviewer_KytH · 2025-02-20

**Summary Of Contributions:**

The paper attempts to investigate recent methods of conformal prediction for graph-structured data, and specifically for the node classification task. Particular focus is placed on two key papers in that direction, CFGNN (Huang et al. 2023) and DAPS (Zargarbashi et al. 2023), and to a lesser extent NAPS (Clarkson 2023). Some background is provided, and then more general conformal scoring approaches for classification are discussed, including class-wise quantiles and randomization in APS, and some additional experiments w.r.t. to the graph-based methods are done. Finally, CFGNN's training time is improved by batching and some general comparison of methods is done.

**Audience:**

No

**Broader Impact Concerns:**

I do not believe there are any ethical concerns with this work.

**Claims And Evidence:**

No

**Requested Changes:**

Personally I think the work needs a substantial overhaul and more extensive comparison experiments in order to qualify as an interesting benchmarking paper.

On a more detailed note, some remarks on notation in the background:
- The miscoverage level $\alpha$ is usually restricted to $(0,1)$ rather than $[0,1]$ in order to avoid edge cases and thus similarly $1-\alpha \in (0,1)$
- $x_{n+1}$ should be specified as being a previously unseen test sample
- Another implicit assumption is that the scoring function $s$ is applied point-wise to each sample in order to ensure that data exchangeability translates to exchangeability of the nonconformity scores (which are used in practice to construct the quantile)
- In Thm. 2.1 the upper coverage bound $1-\alpha+\frac{1}{n+1}$ is exact (i.e. $\geq$ not >) and requires an additional assumption on uniqueness of the scores
- The set $C_{\alpha}(X)$ is denoted using what I assume is a random variable $X$, but is then defined using observation $x$
- Even in full CP the scores are still computed point-wise or per sample but simply include a predictor retrained on the dataset with candidate $(x_{n+1}, y)$ so I don't see why the score function there is deemed "more expensive as it maps $\mathcal{X}^k$ ..."
- The classification task is defined as the map $(\mathbf{X}, \mathbf{A}, v) \mapsto y_v$, but perhaps given the notation before $(x_v, \mathbf{A}) \mapsto y_v$ is more suitable (since the map is now specific -> specific rather than general -> specific)
- The distinction between transductive and inductive (especially since used differently in standard conformal literature) for the graph setting should be discussed
- for the class-wise TPS quantiles $\hat{q}(\alpha, y_j)$ the way they are defined an in particular how the subsetted score set is defined seems mathematically confusing. Also I assume $j$ is some targeted class label $j = 1, \dots, K$?

**Strengths And Weaknesses:**

Some of CFGNN's and DAPS' design choices -- e.g. in terms of data splitting or employed conformal scoring methods -- are thoroughly discussed, and the authors seem to have spent time understanding the methods well. However, the overall obtained insights and interest in contributions remain unclear to me. In sec. 2 the background follows the referenced works (while making some notational mistakes, see below), and Thm 2.2. is essentially a direct copy from (Huang et al. 2023), which is fine.

In sec. 3 some time is spent discussing experimental design including different data partitioning schemes, which can also be inferred from the respective papers directly. Then TPS (Sadinle et al. 2019) as a conformal scoring method is discussed, and its class-wise version is introduced following (Sadinle et al. 2019). The approach is motivated as more 'adaptive', but it remains unclear why TPS or TPS-classwise are of particular interest in the first place for graph data, since existing approaches comment on TPS and opt to use improved APS (Romano et al. 2020) or RAPS (Angelopoulos et al. 2021) instead. In particular, the comparison between TPS (which provides marginal guarantees) and TPS-classwise (which provides stronger class- or partition-conditional guarantees, also called mondrian CP) seems insufficient, and the fact that they address different coverage targets and thus are not necessarily directly comparable is not properly discussed. In Fig. 1, it is unclear to me what the x-axis symbolizes -- is the visible coverage distribution across class labels? Then I would expect TPS-classwise to exactly match target coverage for every class, and TPS should still marginally achieve target coverage. The fact that for the considered LC data split only sample counts of {$10, 20, 40, 80$} are considered *per class* also casts doubt on the robustness of obtained class-wise results.

Next, an extended section discusses the randomization or tie-breaking component of APS, and how APS with tie-breaking provides more efficient prediction sets than without. The insight is fairly trivial and has been discussed before in both the APS and RAPS papers, since it simply suggests a randomized decision on inclusion of one additional label in the set (versus inclusion every time), and thus naturally will sometimes produce label sets with one less label. The inclusion of an own theorem and proof seems unnecessary for this intuition. More specifically relating to graph data, it is again unclear why APS and the randomization step are of such particular importance, except that randomization expectedly improves some results and seems to have been ignored by CFGNN. Yet, the standard APS approach includes randomization and thus trivially provides the obtained benefits, if followed correctly.

After that, training improvements for CFGNN in terms of runtime are suggested by introducing batching and caching. Again, not really interesting from a CP perspective and the training changes do not provide any other benefits (e.g. performance improvements), so I struggle to call this an "insight" for either CP or the graph application.

Finally, an overall comparison of methods is done across datasets. Personally, I find Fig. 7 the most interesting results and closest in line with claims on "benchmarking". However, the different employed scoring methods are not properly explained and it is hard to draw personal conclusions. Furthermore, the considered datasets (aside from Flickr) are all not novel for this task, and have been used in most parts both by (Huang et al. 2023) and (Zargarbashi et al. 2023). So technically, the obtained insights are quite limited. I would have expected extensive comparisons across more datasets and settings of this sort in the paper, e.g. to the scope of (Dheur and Taieb 2023, https://proceedings.mlr.press/v202/dheur23a/dheur23a.pdf) who did a proper benchmarking study for calibration. The related works section afterwards is extremely limited and seems to only focus on the three aforementioned papers, whereas I would expect a thorough literature search and discussion of any related papers.

Overall, the different proposed insights are either very limited in their insight or seemingly unrelated to conformal prediction for graph data. (Huang et al. 2023) explore both classification and regression tasks, while (Zargarbashi et al. 2023) do in-depth experiments for classification across most of the datasets considered here. This paper proposing to benchmark and draw new insights is experimentally more limited than either one of the approaches, and it is hard to draw a conclusive summary of obtained interesting insights.

---

> ### Author Response · Authors · 2025-03-01
> **Response to Reviewer [1/2]**
>
> We thank the reviewer for taking the time to review our paper.
>
>
> > [W1] TPS vs TPS-Classwise
>
> >> [W1a] "...why TPS or TPS-classwise are of particular interest in the first place for graph data..."
>
> Clearly establishing the improvements made by different graph conformal prediction methods requires using TPS as a baseline, as it is provably optimal in a non-graph setting. However, it is not necessarily adaptive, so we add TPS-classwise as a strong adaptive baseline.
>
> >> [W1b] "...the comparison between TPS ... and TPS-classwise ... seems insufficient, and the fact that they address different coverage targets and thus are not necessarily directly comparable is not properly discussed."
>
> Both TPS and TPS-Classwise achieve the $1 - \\alpha$ coverage like other CP methods such as APS and RAPS. The main distinction is that TPS-Classwise also guarantees that $\\text{Pr}(y_{n + 1}\\in C_{TPS}({x}\_{n + 1}) \\mid y\_{n + 1} = y) \\geq 1 - \\alpha,~\\forall y\\in\\mathcal{Y}$.
>
> >> [W1c] "In Fig. 1, it is unclear to me what the x-axis symbolizes -- is the visible coverage distribution across class labels? Then I would expect TPS-classwise to match target coverage for every class exactly, and TPS should still marginally achieve target coverage. The fact that for the considered LC data split only sample counts of $\\{10, 20, 40, 80\\}$ are considered _per class_ also casts doubt on the robustness of obtained class-wise results."
>
> The x-axis refers to the label stratified coverage for the test set, which is defined as the average of the coverages achieved for each class. It is well-known that coverage follows a (beta) distribution across multiple samples, hence we would not expect exact (1- $\\alpha$) coverages. The variance in the distribution of coverages is higher for smaller sample counts, so using $10$ samples per class (which is lower than what other works in the literature use e.g. DAPS uses 20 per class) helps establish robustness. Furthermore, we also evaluate using full-split (FS) where we do not control for the number of nodes per class and achieve similar (or even strong) results as seen in Figure 1a (left) and 1b (left).
>
> > [W2] APS vs APS Randomized
>
> > >[W2a] "More specifically relating to graph data, it is again unclear why APS and the randomization step are of such particular importance..."
>
> Like TPS and TPS-Classwise, APS and APS Randomized are important as they are commonly used baselines in the CP for classification literature -- both graph and non-graph. These methods provide a good balance of efficiency and adaptability which other methods can compare against. Furthermore, APS is foundational to certain graph methods such as DAPS and CFGNN's efficiency-based loss, so, as the reviewer acknowledged, careful considerations with APS can result in stark improvements for these methods as shown in Figure 5 in our manuscript.
>
> > >[W2b] "... insight is fairly trivial and has been discussed before in both the APS and RAPS papers...and thus naturally will sometimes produce label sets with one less label."
>
> While this insight has been stated for APS and APS randomized (in the general setting), we could not find clear proof in the existing literature. Further, our empirical results, as shown in Figure 4, demonstrate that using APS randomized can improve efficiency (prediction set size) by more than just one label at test time. This is because unlike in the calibration phase, you do not know the correct label of a test point. Thus, you need to apply randomization for every score. This can also be theoretically shown and we will add that as a corollary to Theorem 3.1. Furthermore, Theorem 3.1 can be used with the calibration set to determine which method will be better at test time, apriori. Lastly, it is essential to make this point, especially in light of recent graph CP efforts (e.g. CFGNN that did not examine APS with randomization as a baseline).
>
> > [W3] "... not really interesting from a CP perspective and the training changes do not provide any other benefits (e.g. performance improvements), so I struggle to call this an "insight" for either CP or the graph application."
>
> The purpose of these improvements is to (a) allow for larger graphs (e.g. ogbn-products scale graphs) to leverage the discussed CP methods (e.g. CFGNN, NAPS) and (b) discuss our library implementation, which is another important facet of our work.
>
> > [W4] "...the considered datasets (aside from Flickr) are all not novel for this task.."
>
> This work focuses on graph datasets and node classification, which is more limited in the number of datasets compared to tabular datasets used in the example paper the reviewer provided. Here is an example of a recent benchmarking-like paper for calibration using graph datasets, [https://openreview.net/pdf?id=lxN59fWoXC](https://openreview.net/pdf?id=lxN59fWoXC). Our work uses a comparably large number and variance of graph datasets (including much larger graphs such as ogbn-products results in the appendix).

---

> ### Author Response · Authors · 2025-03-01
> **Response to Reviewer [2/2]**
>
> > [RC1] "...some remarks on notation in the background:"
>
> We thank the reviewer for these notational remarks and have made the necessary fixes in the updated version. There are some remarks that we would like to clarify which we numbered below.
>
> - [6] In full conformal prediction, for computing the point-wise score, the score function has $n + 1$ arguments which makes it expensive to compute as compared to split CP, where the score function only has a single argument (i.e. $({x}_i, y_i)$). We will clarify this in the updated version.
>
> - [7] The definition of the node classification task includes all of the necessary components for a GNN which are the feature matrix $X$, adjacency matrix $A$, and the node information $v$, where only giving $(x_v, A)$ would not suffice since you need neighbor feature information.

---

### Review · Reviewer_1Kwg · 2025-02-24

**Summary Of Contributions:**

This paper mainly performed experiments to compare different conformal prediction methods for node classification. It also introduced Theorem 3.1, however, it seems irrelevant to node classification.

**Audience:**

Yes

**Claims And Evidence:**

Yes

**Requested Changes:**

- This paper only studies node classification, therefore, it is better to modify the title to reflect this.
- The comparison of randomized, non-randomized APS, and conformalized GNN is a good contribution. However, I wonder if the authors can add what is special about these comparisons w.r.t. node classification. For example, how is the result different from that for i.i.d. data (e.g., image classification)?
- A crucial aspect of conformal prediction for node classification is the selection of calibration data to satisfy the exchangeability assumption. For example, when given a test set, selecting which nodes as the calibration set can make the exchangeability assumption hold? Should each test node have a unique calibration set? This work can be improved by taking the design choice of the calibration set into consideration.
- Label stratified coverage is not formally defined.

**Strengths And Weaknesses:**

Strength
- Comparison between APS and randomized APS: Rich empirical results verify the advantage of randomized APS over APS.
- Results show the advantage of conformalized training over randomized APS is not that significant, which is a new observation compared to previous work.
- It improved the efficiency of conformalized training to handle larger graphs.

Weakness
- Although the experiments are performed on graph data (node classification). The conformal prediction methods such as APS and randomized APS, and the theorem 3.1 are not specifically designed for graph data.
- The paper is a benchmark paper but it does not contribute a general tool for benchmarking any conformal prediction method for node classification. It does not contribute new datasets either. I wonder if the significance of the contribution is good enough.

---

> ### Author Response · Authors · 2025-03-01
> **Response to Reviewer**
>
> We thank the reviewer for taking the time to review our paper.
>
> > [W1] "The conformal prediction methods such as APS and randomized APS, and the theorem 3.1 are not specifically designed for graph data."
>
> The reviewer is correct in noting that Theorem 3.1 is more generally applicable to CP as a whole, rather than just graph CP which was the focus of our work. We include analysis of non-graph methods, such as APS, as they are common baselines and or are fundamental to graph CP methodologies (e.g. in CFGNN's efficiency loss, DAPS). Thus, it is important to note when these methods perform best to ensure a fair and clear understanding of the literature. The significance of randomization for APS is formally established in Theorem 3.1. While some practitioners may be aware of this, we found no clear proof in the existing literature. It is particularly important to emphasize this, as recent methods, including the state-of-the-art CFGNN, fail to account for randomization when comparing against the baseline APS approach. As shown in Figure 5, APS without randomization can artificially inflate the benefits of CFGNN.
>
> > [W2] "The paper is a benchmark paper but it does not contribute a general tool for benchmarking any conformal prediction method for node classification. It does not contribute new datasets either"
>
> In addition to the analysis and results, another contribution of this work is our graph CP Python library which provides scalable implementations that practitioners can use to evaluate the discussed CP methods with other methods that may be developed. This library is part of the supplemental material and will be made public if the paper is accepted.
>
> > > [RC1] This paper only studies node classification, therefore, it is better to modify the title to reflect this.
>
> We can change the title to specify node classification, but we would like to note that the majority of the existing work in the graph conformal prediction literature is for node classification.
>
> > > [RC2] The comparison of randomized, non-randomized APS, and conformalized GNN is a good contribution. However, I wonder if the authors can add what is special about these comparisons w.r.t. node classification. For example, how is the result different from that for i.i.d. data (e.g., image classification)?
>
> In image data, RAPS utilizes the fact that typical image classification datasets have a large number of classes (e.g. ImageNet has 1000 classes). CFGNN and DAPS aim to utilize the structure and homophily of the graph and aim to improve upon methods that ignore the graph structure (e.g. APS, TPS). Our work establishes the specific observations about their setup under which they improve. We will also clarify this by summarizing these in the introduction and background.
>
> > > [RC3] A crucial aspect of conformal prediction for node classification is the selection of calibration data to satisfy the exchangeability assumption. For example, when given a test set, selecting which nodes as the calibration set can make the exchangeability assumption hold? Should each test node have a unique calibration set? This work can be improved by taking the design choice of the calibration set into consideration.
>
> Perhaps, there is a misunderstanding, for exchangeability, the calibration must be drawn from the union of the calibration and test points in non-graph settings. Theorem 2.2 allows us to also sample exchangeably in the graph setting because of the equivariance to structure across the calibration and test nodes. Conditional sampling may violate the exchangeability assumption.
>
> > > [RC4] Label stratified coverage is not formally defined.
>
> We are not sure if the reviewer is referring to the metric formulation or the coverage guarantee of label stratified coverage. If the reviewer is asking about the metric, then it is formulated as $\\frac{1}{|\\mathcal{D}\_{test}|}\\sum\_{i\\in\\mathcal{D}\_{test}} \\Big(\\frac{1}{|\\mathcal{Y}|}\\sum\_{y\\in\\mathcal{Y}}\\mathbf{1}[y\_{i}\\in C({x}\_i), y\_{i} = y]\\Big)$. If it is the coverage guarantee, then it is formulated as $\\text{Pr}(y\_{n + 1}\\in C({x}\_{n + 1}) \\mid y\_{n + 1} = y) \\geq 1 - \\alpha,~\\forall y\\in\\mathcal{Y}$. We will clarify this in the updated version.

---

> ### Comment · Reviewer_1Kwg · 2025-03-04
>
> > We can change the title to specify node classification, but we would like to note that the majority of the existing work in the graph conformal prediction literature is for node classification.
>
> There exist papers like [1,2] which work on conformal link prediction.
>
> [1] Zhao, Tianyi, Jian Kang, and Lu Cheng. "Conformalized link prediction on graph neural networks." Proceedings of the 30th ACM SIGKDD Conference on Knowledge Discovery and Data Mining. 2024.
> [2] Blanchard, Gilles, et al. "FDR control and FDP bounds for conformal link prediction." arXiv preprint arXiv:2404.02542 (2024).
>
> > In image data, RAPS utilizes the fact that typical image classification datasets have a large number of classes (e.g. ImageNet has 1000 classes). CFGNN and DAPS aim to utilize the structure and homophily of the graph and aim to improve upon methods that ignore the graph structure (e.g. APS, TPS). Our work establishes the specific observations about their setup under which they improve. We will also clarify this by summarizing these in the introduction and background.
>
> Can the authors please specify, in terms of methodology, how CFGNN and DAPS are special w.r.t. graph data, especially compared with [3]? For example, is there any novel methodology introduced to improve the efficiency?
>
> [3] Stutz, David, et al. "Learning Optimal Conformal Classifiers." International Conference on Learning Representations.
>
> > Perhaps, there is a misunderstanding, for exchangeability, the calibration must be drawn from the union of the calibration and test points in non-graph settings. Theorem 2.2 allows us to also sample exchangeably in the graph setting because of the equivariance to structure across the calibration and test nodes. Conditional sampling may violate the exchangeability assumption.
>
> Theorem 2.2 is just one way to make exchangeability hold in graph data, which is based on the transductive setting and the assumption that calibration and test sets are exchangeable. That is why the paper should come with **transductive node classification** in its title or expand its scope to consider how exchangeability can hold in inductive settings or other graph learning problems (e.g., graph classification and link prediction).

---

> ### Author Response · Authors · 2025-03-08
> **Response to Reviewer**
>
> Thank you for your comments. Responses follow.
>
> > There exist papers like [1,2] which work on conformal link prediction.
>
> While a couple of recent works (the first vetted publications were in June and Aug 2024)  have focused on conformal link prediction, there is simply not enough work to warrant a full benchmarking study on this task yet. [1] appeared at SIGKDD last year and we do cite it.  [2] is not yet vetted to our knowledge, although it appears to build on [3], which we also had cited in the related work section. The method in [3] appeared in a Springer journal (Spanish Society of Statistics and OR) and is tailored towards controlling FDR. Furthermore, [3] primarily evaluates the method on synthetic data. The one real data set they report results on has 81 nodes and 221 edges - this dataset is too small and below the recommended number of data points for evaluating conformal prediction methods considered (as noted by [4]).
>
> [1] Zhao, Tianyi, Jian Kang, and Lu Cheng. "Conformalized link prediction on graph neural networks." Proceedings of the 30th ACM SIGKDD Conference on Knowledge Discovery and Data Mining. 2024.
>
> [2] Blanchard, Gilles, et al. "FDR control and FDP bounds for conformal link prediction." arXiv preprint arXiv:2404.02542 (2024).
>
> [3] Marandon, Ariane "Conformal link prediction for false discovery rate control", J. of Spanish Society of Statistics and OR, 2024, June 2024.
>
> [4] Angelopoulos, A. N., Bates, S. A Gentle Introduction to Conformal Prediction and Distribution-Free Uncertainty Quantification, Dec 2022.
>
> > Can the authors please specify, in terms of methodology, how CFGNN and DAPS are special w.r.t. graph data, especially compared with [5]? For example, is there any novel methodology introduced to improve the efficiency?
>
> CFGNN shares some similarities to [5] in that it uses a secondary model and a specialized loss function to learn the conformal predictor. As noted in [6], the key difference is that CFGNN uses GNN to utilize network homophily and graph structure to improve efficiency, whereas [5] focuses on image data. We include CFGNN in our study as it is considered state-of-the-art for graph conformal prediction for transductive node classification. We have included [5] in our discussion of CFGNN in Section 2.2.
>
> DAPS differs from CFGNN and [5] in that it doesn’t train a secondary model but instead applies a one-step graph diffusion update on the APS scores, utilizing the graph structure and homophily, similar to CFGNN.
>
> [5] Stutz, David, et al. "Learning Optimal Conformal Classifiers." International Conference on Learning Representations.
>
> [6] Huang, Kexin, Ying Jin, Emmanuel Candes, and Jure Leskovec. "Uncertainty quantification over graph with conformalized graph neural networks." Advances in Neural Information Processing Systems 36 (2024).
>
> > Theorem 2.2 is just one way to make exchangeability hold in graph data, which is based on the transductive setting and the assumption that calibration and test sets are exchangeable. That is why the paper should come with transductive node classification in its title or expand its scope to consider how exchangeability can hold in inductive settings or other graph learning problems (e.g., graph classification and link prediction).
>
> We have updated the title of our work to “**Conformal Prediction: A Theoretical Note and Benchmarking Transductive Node Classification in Graphs**”. This title emphasizes that we are focusing on transductive node classification, but still emphasizes the important and general applicability of our key theoretical insight (i.e., Theorem 2.4 in the updated manuscript). Thank you for the suggestion!
> Additionally, we have moved our discussions of link prediction and other tasks to the appendix to emphasize our focus on the transductive node classification task in the main body.

---

### Review · Reviewer_TeCC · 2025-02-25

**Summary Of Contributions:**

This paper studies a few design choices involved when using conformal prediction for graph classification. The authors review some of the literature on conformal prediction (including TPS, adaptive TPS, and randomized and non-randomized APS). They then present a general theorem (not specific to graphs) analyzing in which circumstances one nonconformity score function produces more efficient results than another. Additionally, they discuss the conformalized GNN (CFGNN) from Huang et al. (2023) and provide an implementation that is more efficient and allows varying the nonconformity score function used, and also discuss a  variant DTPS of the previously proposed DAPS algorithm (H. Zargarbashi et al. 2023). These results are each accompanied by empirical findings on graph classification datasets, focusing on the efficiency and label-stratified coverage of each method.

**Audience:**

No

**Broader Impact Concerns:**

No broader impact concerns.

**Claims And Evidence:**

No

**Requested Changes:**

## Fix clarity and possible correctness issues in Theorem 3.1 [W1]
I found Theorem 3.1 difficult to follow and am not sure the definitions make sense. I think this should be clarified and fixed if necessary.

In particular, before Theorem 3.1 the authors define:

> Let $\alpha\_c^A$ be the significance level of incorrect labels being in the prediction set, i.e
>
> $$
> 1-\alpha\_c^A \leq \Pr[A(\mathbf{x}\_{n+1}, y\_{n+1}') < q\_A] < 1-\alpha\_c^A + \frac{1}{n+1}
> $$
> $$
> \text{ for } y\_{n+1}' \in \\{1,2,\ldots,K\\}\backslash\\{y\_{n+1}\\}
> $$

It's not clear to me that this is a valid definition of $\alpha_c^A$. In the multi-class case, $y\_{n+1}'$ is not a single value but instead an element of a set of values. But then it is not obvious that there is a single value of $\alpha_c^A$ for which the desired inequality holds. Additionally, it does not seem trivial to "invert" the result of Theorem 2.1, which shows how to compute $q$ from $\alpha$ but not the reverse. (In fact I'm not sure if Theorem 2.1 makes sense if $y\_{n+1}'$ is an arbitrary element of a set rather than a fixed label.)

A similar issue arises in the derivation after "For $K$ classes", where we take a sum of probabilities for each value of $y\_{i}'$ and turn that into $(K-1)\Pr[y_i' \in C_A^{n+1}]$. This seems incorrect, because at this point $y_i'$ is not bound and could take multiple different values.

Beyond this, the notation $C\_A^{n+1}$ also does not seem to be explicitly defined, although it may be equivalent to $C\_A(x_{n+1})$?

## Fix issue with "Label-Count Sample Partitioning" [W2]
Page 4 describes "Label-Count (LC) Sample Partitioning" as:
> The data is split to ensure an equal number of samples for each class label is present in Dtrain, Dvalid, and Dcalib. The remaining nodes are Dtest.

However, this seems like it would break exchangeability. If data is split to ensure some property is true about Dcalib, this implies that Dcalib and Dtest are chosen via different processes, so conformal prediction results should not apply.

This also seems to contradict the statement in Section 2.1 that "The specific Vcalib and Vtest are randomly sampled from Vcalib ∪ Vtest."

I think this should be fixed.

## Clarify the overall findings and takeaways [W3, W4]
As it currently stands, the paper is not very clear about what the contributions, main claims, and takeaways are. The abstract and introduction are quite vague, and the organization of the paper does not make it clear what parts of the paper are the main contributions. Additionally, many of the contributions seem somewhat unrelated, and I am not sure if there is a central idea that ties them together. Finally, although the set of experiments seems fairly thorough, the paper does not seem to have much discussion about the interpretation of the results, and so it is not clear to me what these results actually tell us about the design choices and tradeoffs involved.

Overall I think this paper would be improved if it was more explicit about these and made some more specific claims or recommendations about what the results mean. Although less important than the correctness issues above, I think this is also important to address.

---

Beyond these main issues, I also found a few more minor issues and had a few questions:
- Section 2.1 denotes dataset subsets with $V\_{\text{split}}$, but section 3 denotes them with $D\_{\text{split}}$. Should these match?
- Section 2.1 states "Next, we discuss the different settings for node classification in graphs" but as far as I can tell it only discusses a single setting.
- $V\_{\text{test}}$ appears twice in the second line of "Transductive setting", I assume one of these should be "train".
- It's a bit confusing to me that a *smaller* value of "efficiency" means that the method is *more* efficient rather than less. Is this common in the literature? If not, could you call it "inefficiency" or similar instead?

**Strengths And Weaknesses:**

## Strengths

- **S1**: The authors do a good job of summarizing prior work and explaining conformal prediction.
- **S2**: The paper includes a number of experiments where the nonconformity score and sample partitioning strategy are varied systematically, which seems valuable for understanding the tradeoffs between the methods.
- **S3**: The authors include code to reproduce their experiments, which also includes their efficient implementation of CFGNN.

## Weaknesses

- **W1**: I found Theorem 3.1 difficult to understand and am not sure that it is correct. See my comments in the "requested changes" section below.
- **W2**: The "Label-Count Sample Partitioning" strategy described in Section 3 seems to break exchangeability requirement from Section 2.1. I'm not sure if this is a fundamental issue or an error in describing the strategy.
- **W3**: I do not have a clear sense of what the major claims and takeaways are from this work. I'm not sure what the empirical findings mean in practice (other than that there are a lot of choices of algorithm and that they have various dataset-dependent tradeoffs).
- **W4**: The contributions in this work seem somewhat unrelated to each other,  making the paper feel less cohesive. Most of the comparisons between nonconformity scores seem to be properties of conformal prediction in general, rather than being graph specific. On the other hand, the efficient implementation of the CFGNN approach is very specific to a particular method. I don't have a clear sense of who the intended audience is.

---

> ### Author Response · Authors · 2025-03-01
> **Response to Reviewer**
>
> We thank the reviewer for taking the time to review our paper.
>
> > [W1] Fix clarity and possible correctness issues in Theorem 3.1
>
> To improve clarity for Theorem 3.1, we have updated the notation and exposition in the manuscript with a more detailed description of how $\\alpha_{c}^{A}$ is defined, including how to "invert the result of Theorem 2.1" to calculate $\\alpha_{c}^{A}$ given $\\hat{q}_A$.
>
> > [W2] Fix issue with "Label-Count Sample Partitioning"
>
> Indeed this is a biased sampling process, but we include it in our evaluation to benchmark for this strategy which has been used to evaluate graph conformal prediction methods, most notably [1, 2]. We do not claim anything about the exchangeability under this setting. We will make this fact explicit in the section describing this partitioning strategy in the updated manuscript.
>
> [1] Soroush H. Zargarbashi, Simone Antonelli, and Aleksandar Bojchevski. Conformal prediction sets for graph neural networks.
>
> [2] Soroush H. Zargarbashi, Mohammad Sadegh Akhondzadeh, and Aleksandar Bojchevski. Robust Yet Efficient Conformal Prediction Sets
>
> > [W3, W4] Clarify the overall findings and takeaways
>
> >> [a] "...not very clear about what the contributions, main claims, and takeaways are. The abstract and introduction are quite vague..."
>
> We will update the final paragraph of the introduction to be a separate subsection titled **Key Contributions** so that our contributions and takeaways are made concrete. In this section, we will note the following 3 contributions:
>
> 1. We perform a comprehensive analysis of the existing graph split-CP literature (e.g. CFGNN, DAPS) for node classification. This analysis aims to understand the choices made in method implementation (i.e., scalability) and method evaluation (i.e., dataset preparation, and baseline selection). We further expand our analysis to include the broadly applicable CP methods the literature discusses and uses like TPS and APS.
>
> 2. We provide a theorem for evaluating different conformal prediction methods in terms of efficiency (i.e., set size) _prior_ to deployment.
>
> 3. We created a Python library that captures the different parts of our analysis that practitioners can use when implementing and evaluating conformal prediction for graph data. We further implemented the methods to be scalable for large graphs, particularly for computationally expensive methods like CFGNN.
>
> >> [b] "... not sure if there is a central idea that ties them together."
>
> The focus of this work and the central idea is to benchmark the existing literature in graph conformal prediction, which differs in terms of design choices, implementations, and baseline. In this work, we bring together these differences into a single manuscript and provide recommendations to a practitioner who can apply which we address in point [c] below.
>
> >> [c] "... it is not clear to me what these results actually tell us about the design choices and tradeoffs involved."
>
> The results help us formulate a set of recommendations for practitioners to use when using different conformal prediction methods on graphs, which we also will summarize in bullet points in Section 3.4 of the updated manuscript.
>
> > > [RC1] "...few more minor issues and had a few questions"
>
> - They are different but are closely related. $\\mathcal{V}$ is denoting the vertices in the graph, while $\\mathcal{D}$ is denoting the $({x}, y)$ data point.
> - We thank the reviewer for pointing this out. The updated manuscript will also include a description of the inductive node classification setting.
> - This will be fixed in the updated version.
> - Both "efficiency" and "inefficiency" are commonly used in the literature.

---

> > ### Comment · Reviewer_TeCC · 2025-03-04
> > **Discussion**
> >
> > Thank you for the reply.
> >
> > **Contributions and overall takeaways:**
> >
> > I'm still a bit confused about the claim that you "perform a comprehensive analysis of the existing graph split-CP literature". I'm not sure if this is just an issue with the organization of the paper, but after reading it it is still unclear to me what you mean by "comprehensive analysis". Section 3 seems to be organized as a sequence of individual pieces of background information and specific empirical considerations about specific methods rather than a comprehensive analysis of the full literature, and I was expecting something more systematic.
> >
> > As a specific example, the introduction states that you aim to "understand the choices" made in "i.e., dataset preparation". In Section 3 you list two ways that previous work has partitioned samples, but I don't feel like I understand why these choices are made or what this tells us about the previous literature. (Apart from the fact that LC seems to violate exchangeablility, as I stated above, which means it's not clear how any of the numbers for LC should be interpreted anyways?)
> >
> > I think overall the paper would be improved by being more focused on a specific set of questions, and organizing the paper around those questions. "Performing a comprehensive analysis" is vague enough that it's not clear to me how the individual findings contribute. I do appreciate the new summary in Section 3.4, which gives things a bit more structure than in the previous version, but this comes at the end of an 8-page section and it isn't clear until then what questions you are trying to answer or what your experiments are showing. (One possibility: maybe you could split Section 3 into four sections: the new baselines you are proposing / highlighting, the overall empirical "benchmarking" results, the theoretical analyses, and the implementation efficiency concerns. Then each section could clearly frame the specific question that is answered by that section without readers getting confused.)
> >
> > **Theorem 3.1 and Appendix B:** Thank you for expanding on the definitions in section 3.2, I better understand what $C^i_A$ and $y_i^R$ are now. However, I am still having some trouble understanding the derivation. In particular Lemma B.1 doesn't make sense to me as written, because the LHS is a number and the RHS is a random variable. Unless this probability is implicitly a conditional probability given some not-explicitly-stated event? This issue also affects Lemma 3.1 and Equation 7; $\alpha\_c^A$ is a random variable but the probability in the middle is just a number. Are you implicitly conditioning on something?
> >
> > **Dataset subsets:** Section 2.1 seems to use both $\mathcal{D}$ and $\mathcal{V}$ without explaining how they are related (are the data points in $\mathcal{D}\_\text{train}$ just the vertices in $\mathcal{V}\_\text{train}$ with their labels)? I'd suggest either clarifying the relationship between $\mathcal{D}$ and $\mathcal{V}$ or switching to use only $\mathcal{V}$ starting in section 2.1.

---

> > > ### Author Response · Authors · 2025-03-08
> > > **Response to Reviewer**
> > >
> > > > Contributions and overall takeaways
> > >
> > > We appreciate the reviewer’s suggestions for reorganization and are continuously improving the manuscript. We updated the manuscript as follows:
> > >
> > > 1. For clarity, we moved the theoretical analysis surrounding Theorem 3.1 (now Theorem 2.4) to precede the empirical results under the “Conformal Prediction” section (i.e. Section 2).
> > > 2. We added a new section titled “Conformal Prediction Methods”. The section introduces and elaborates on all the CP methods discussed in our work including DAPS, DTPS, NAPS, and RAPS. It is placed in Section 2.2 to minimize fragmentation.
> > > 3. We restructure Section 3 to align with the reviewer’s suggestion (with the exception that the theoretical analysis is now moved to Section 2 before the empirical analysis) to facilitate improved exposition. We also include several key research questions at the start of Section 3 as suggested by the reviewer– these are addressed in subsequent subsections.
> > >
> > > The choices made in dataset preparation align with the types of classification tasks researchers are interested in evaluating. The FS split style is reasonable as a larger training and validation set is available for development like in standard supervised node classification. For the semi-supervised setting – which [1] was interested in– the LC split is more natural as the labeled data available during development is less. To reiterate, we agree that the LC split may violate exchangeability, thus not providing the same theoretical guarantees; however, since it has been used in the literature, we include it in our work for completeness (see Dataset Splits and Training under Label-Count (LC) Sample Partitioning).
> > >
> > > [1] Soroush H. Zargarbashi, Simone Antonelli, and Aleksandar Bojchevski. Conformal prediction sets for graph neural networks. Proceedings of the 40th International Conference on Machine Learning, volume 202 of Proceedings of Machine Learning Research
> > >
> > > > Theorem 3.1 and Appendix B
> > >
> > > We thank the reviewer for making this comment. The proof, and many other proofs in the CP literature, implicitly assumes conditioning over $Z_1, \dots, Z_n$ (i.e. the calibration set), when making probabilistic statements about an unknown test point ($Z_{n + 1}$). We have updated the manuscript to clarify and simplify the statement and proof of Lemma B.1 and its application to Lemma 2.3.
> > >
> > > > Dataset subsets
> > >
> > > Yes, the distinction between $\mathcal{D}$ (data points) and $\mathcal{V}$ (vertices) is intentional. We will update the manuscript to clarify this distinction further.

---

### Author Response · Authors · 2025-03-01
**Overview Response to Reviewers and Action Editors**

We thank all the reviewers for taking the time to review our paper and for providing us with helpful feedback to improve the presentation of our work. Below we clarify some of the presentation updates we made as well as emphasize the relevance of certain parts of our work.

## Presentation Updates

Based on the feedback from all the reviewers, we have substantially tightened the writing surrounding Theorem 3.1, the contributions, and the important key insights to be more precise. Specific changes include:

- **Theorem 3.1:** We expand upon the definitions and notations for Theorem 3.1, particularly with $\\alpha_{c}^{A}$, and provide a lemma for how it can be computed using the calibration set. We further clarify how the incorrect labels and the selection of an incorrect label are exchangeable which allows us to $\\alpha_{c}^A$. The proof of Theorem 3.1 has been updated based on the feedback and moved to an appendix.
- **Key Contributions:** We clarify the key contributions of the work and the goals in the introduction to be:

  1. We perform a comprehensive analysis of the existing graph split-CP literature (e.g. CFGNN, DAPS) for node classification. This analysis aims to understand the choices made in method implementation (i.e., scalability) and method evaluation (i.e., dataset preparation, and baseline selection). We further expand our analysis to include the broadly applicable CP methods the literature discusses and uses like TPS and APS.

  2. We provide a theorem for evaluating different conformal prediction methods in terms of efficiency (i.e., set size) _prior_ to deployment.

  3. We created a Python library that captures the different parts of our analysis that practitioners can use when implementing and evaluating conformal prediction for graph data. We further implemented the methods to be scalable for large graphs, particularly for computationally expensive methods like CFGNN.

- **Section 3.4 Rewrite:** We rewrote Section 3.4 to resemble an executive summary of all the key insights that we make in this paper as well as offer some guidelines at the end of what practitioners should do if they are using graph CP. This section includes some of the main takeaways from the discussion in the earlier sections 3.1-3.3.


## Relevance of Theorem 3.1
Theorem 3.1 formalizes the importance of randomization for APS (in a general setting, not limited to graph CP). While some practitioners may know this, we could not find clear proof in the existing literature. Indeed, it is essential to make this point, especially in light of recent efforts (including the state-of-the-art graph CP - CFGNN that did not examine APS with randomization as a baseline). Furthermore, the theorem's utility comes from the fact that $\\alpha_{c}^{A}$ can be computed using the calibration set, allowing a practitioner to determine which of two methods is more efficient (i.e. smaller set size) _prior to_ deployment. We will also provide a corollary of Theorem 3.1, which provably shows that the efficiency improvements with APS Randomized are not limited to just a single label--a fact which seems to have caused some confusion-- but instead, for large calibration sets tends to $(K - 1)(\\alpha\_{c}^{A} - \\alpha\_{c}^{\\tilde{A}})$, where $A$ is the more efficient method and $K$ is the number of classes.

## Relevance of APS and TPS

While our work focuses on conformal prediction for node classification, we include a discussion of TPS and APS (and variants) because of their relevance as baselines in the literature. It is important to highlight TPS as it is provably optimal in terms of efficiency (prediction set size) for non-graph data, though it is not adaptive, which is why we further discuss TPS-Classwise. It is also important to highlight APS and APS randomization as it is the most popular adaptive baseline method, and also because recent graph approaches, including the state-of-the-art CFGNN, do not consider randomization.

Again, we thank the reviewers for their detailed reviews and look forward to further discussion regarding our paper as we continuously improve the presentation of our work.

---

> ### Author Response · Authors · 2025-03-08
> **Overview Response to Reviewers and Action Editors**
>
> We sincerely appreciate the reviewers’ comments and suggestions, which have helped enhance our manuscript.  We are continuously improving the manuscript and made the following changes:
>
> 1. For clarity, we moved the theoretical analysis surrounding the previous Theorem 3.1 (now Theorem 2.4) to precede the empirical results under the “Conformal Prediction” section (i.e. Section 2).
> 2. We added a new section titled “Conformal Prediction Methods”. The section introduces and elaborates on all the CP methods discussed in our work. It is placed in Section 2.2 to minimize fragmentation. Additionally, the section covers some methods (e.g. DAPS, DTPS, NAPS, and RAPS) mentioned in the Overall Results.
> 3. In response to Reviewer TeCC’s suggestion, we restructure Section 3 to align with the reviewer’s suggestion (with the exception that the theoretical analysis is now moved to Section 2 before the empirical analysis) to facilitate improved exposition. We also include several key research questions at the start of Section 3 as suggested by the reviewer– these are addressed in subsequent subsections.
> 4. In response to Reviewer 1Kwg’s suggestion
>      - We updated the manuscript title to “**Conformal Prediction: A Theoretical Note and Benchmarking Transductive Node Classification in Graphs**” as it better reflects our focus on transductive node classification.
>      - We moved the discussion on conformal link prediction and related areas to the appendix to reflect the focus on transductive node classification.
> We have simplified and made the theorem and proof surrounding Lemma 2.3 and the supporting Lemma B.1.
>
> Once again, we thank the reviewers for their detailed reviews and look forward to further discussion regarding our paper as we continue to refine the presentation of our work.

---

### Decision · Action_Editor_V4kc · 2025-04-14

**Recommendation:** Accept as is

**Comment:**

The paper was reviewed by four reviewers [1Kwg,JtiA,KytH,TeCC]. They appreciate aspects of the paper, such as summarizing prior work and explaining conformal prediction [TeCC], thorough discussion of some design choices [KytH], a theoretical result of sufficient conditions for producing an improved prediction set [JtiA], and experiments helping to understand tradeoffs between methods [TeCC] and showing advantages of randomized adaptive prediction sets [1Kwg,JtiA].

However, originally, the reviewers had strong concerns as well: they identified concerns of vague and unsupported claims [JtiA], correctness issues including for one theorem [TeCC] and suitability of the methods and theorem for graph data [1Kwg]. Reviewers also found the claims unclear [TeCC] and the contributions not cohesive [TeCC] and had concerns of limited insight [KytH], overall significance/amount of the contributions [1Kwg] as a benchmarking paper. More comparison experiments were desired [KytH]. M

Authors provided responses, and there was also in-depth discussion between some reviewers and the authors.

The reviewers appreciated the revision work by the authors, and all four reviewers ultimately leaned towards acceptance, although one remained borderline in their evaluation of the interestingness of the paper:
- The reduction in the scope of the claims was appreciated, and the claims were now considered clearly specified, with adequate supporting discussion, theorem, and results.
- Some concern about organization of the paper, cohesiveness of the claims/takeaways, and thoroughness of the findings remained.
- However, the paper was considered better structured than before.
- The paper was seen to at least reaffirm claims of previous papers and provide some practical advice, and some findings about the baselines challenging previous assumptions were appreciated.

Overall, I lean slightly towards acceptance.

**Audience:**

Reviewers had mixed feelings regarding the interestingness of the paper to the TMLR audience. However, according to reviewers, after revision, the paper was seen to at least reaffirm claims of previous papers and provide some practical advice, and some findings about the baselines challenging previous assumptions were appreciated. Thus, it seems at least some practitioners of graph conformal prediction within the TMLR audience might appreciate the paper.

**Claims And Evidence:**

In the originally submission, the reviewers had strong concerns about the claims and evidence: they identified concerns of vague and unsupported claims [JtiA], correctness issues including for one theorem [TeCC] and suitability of the methods and theorem for graph data [1Kwg]. Reviewers also found the claims unclear [TeCC] and the contributions not cohesive [TeCC] and had concerns of limited insight [KytH], overall significance/amount of the contributions [1Kwg] as a benchmarking paper. More comparison experiments were desired [KytH].

However, after revision by the authors, reviewers felt the situation has improved:
- The reduction in the scope of the claims was appreciated, and the claims were now considered clearly specified, with adequate supporting discussion, theorem, and results.
- Some concern about organization of the paper, cohesiveness of the claims/takeaways, and thoroughness of the findings remained.
- However, the paper was considered better structured than before.

Overall, even though some concerns about cohesiveness of the claims/takeaways, and thoroughness of the findings remain, it seems the current claims may be sufficiently limited in scope and  sufficiently supported.

---

> ### Author Response · Authors · 2025-05-19
> **Thank you!**
>
> We sincerely thank the reviewers and action editors for their thoughtful evaluation of our work and for accepting it without further revisions. In preparing the Camera Ready version, we have incorporated all requested changes from the review process, including a revised title.
>
> -- Authors